# INS-MMBench: A Comprehensive Benchmark for Evaluating LVLMs' Performance in Insurance

## Abstract

Large Vision-Language Models (LVLMs) have demonstrated outstanding performance in various general multimodal applications such as image recognition and visual reasoning, and have also shown promising potential in specialized domains. However, the application potential of LVLMs in the insurance domain—characterized by rich application scenarios and abundant multimodal data—has not been effectively explored. There is no systematic review of multimodal tasks in the insurance domain, nor a benchmark specifically designed to evaluate the capabilities of LVLMs in insurance. This gap hinders the development of LVLMs within the insurance domain. In this paper, we systematically review and distill multimodal tasks for four representative types of insurance: auto insurance, property insurance, health insurance, and agricultural insurance. We propose INS-MMBench, the first comprehensive LVLMs benchmark tailored for the insurance domain. INS-MMBench comprises a total of 8,856 thoroughly designed multiple-choice questions, covering 12 meta-tasks and 22 fundamental tasks. Furthermore, we evaluate multiple representative LVLMs, including closed-source models such as GPT-4o and open-source models like BLIP-2. Our evaluation not only validates the effectiveness of our benchmark but also provides an in-depth performance analysis of current LVLMs on various multimodal tasks in the insurance domain. We hope that INS-MMBench will facilitate the further application of LVLMs in the insurance domain and inspire interdisciplinary development. We will release our dataset and evaluation code.

## 1 Introduction

In recent years, Large Language Models (LLMs) have demonstrated remarkably powerful semantic understanding and conversational capabilities (Wei et al., 2022; Kasneci et al., 2023; Zhao et al., 2023a; Shen et al., 2023; Zhang et al., 2022), profoundly impacting human work and life. Building on this foundation, Large Visual Language Models (LVLMs) have taken a further step by mapping and aligning visual and textual features, enabling the processing and interaction with multimodal data (Bai et al., 2023; Zhu et al., 2023; Wang et al., 2024c; Yin et al., 2023). Researchers have found that LVLMs exhibit exceptional performance in general tasks such as image recognition, document parsing, and OCR processing (Yang et al., 2023; Li et al., 2023b; Xu et al., 2023). Beyond exploring general capabilities, researchers have also begun to apply LVLMs to various specialized domains such as healthcare (Hu et al., 2024; Wang et al., 2024a), autonomous driving (Dewangan et al., 2023; Li et al., 2024b) and social media content analysis (Lyu et al., 2023; Zhang et al., 2024b). By exploring the capabilities of LVLMs in specialized domains through qualitative and quantitative methods, these studies have demonstrated various application potentials.

Insurance, as a discipline encompassing numerous multimodal application scenarios, involves extensive use of multimodal data and computer vision algorithms in its actual operations (Fernando et al., 2022; Sahni et al., 2020; Zhang et al., 2020; Li et al., 2018). This offers vast potential for the integration of LVLMs with the insurance industry. For instance, in auto insurance, analyzing images of damaged vehicles can enable quick assessments and accurate estimations of damage (Mallios et al., 2023). Similarly, in property insurance, analyzing images of buildings can help evaluate

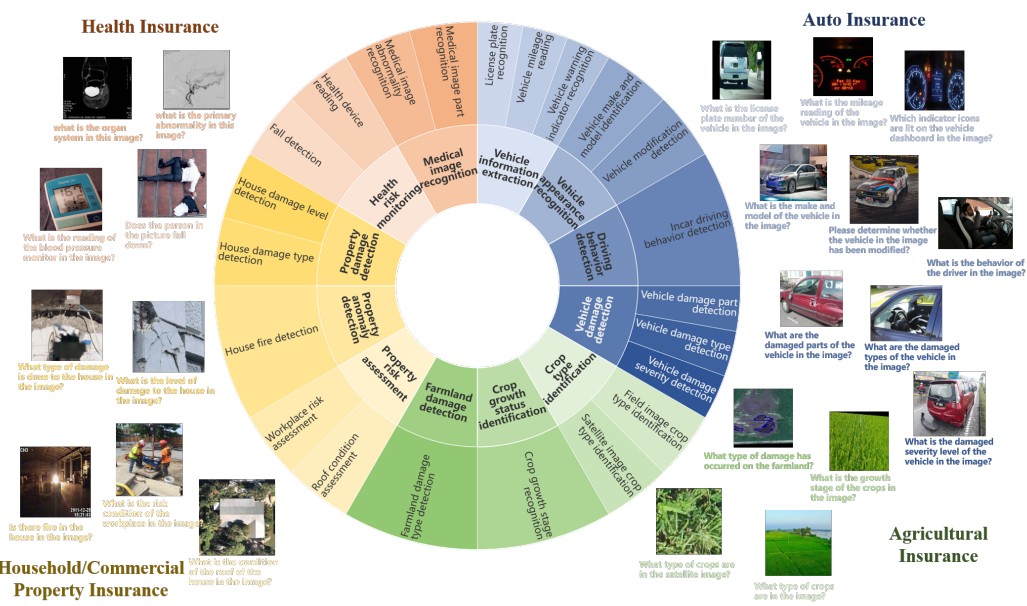

Figure 1: Overview of INS-MMBench. INS-MMBench constructs 12 meta-tasks (represented in the inner circle) and 22 fundamental tasks (represented in the outer circle) across four types of insurance, distinguished by four primary colors: blue, red, yellow, and green. For each fundamental task, we provide an example of image-question pair.

potential risks (Xu et al., 2021). However, existing research (Lin et al., 2024) has only qualitatively analyzed the application of LVLMs in the insurance domain, without systematically organizing related multimodal tasks or constructing domain-specific benchmarks. This has hindered the in-depth evaluation and promotion of LVLMs' capabilities within the insurance domain.

To address this challenge, we introduce INS-MMBench, the first comprehensive LVLMs benchmark for the insurance domain (Figure 1). In our work, we first systematically organize and refine a multimodal task framework across four representative types of insurance: auto, property, health, and agricultural insurance, using a bottom-up hierarchical task definition methodology. Next, we execute a benchmark construction pipeline, including data search, data processing, and question/answer construction. Finally, we propose INS-MMBench, which includes a total of 8,856 thoroughly designed multiple-choice visual questions and images, comprehensively covering 12 meta-tasks and 22 fundamental tasks, spanning key insurance stages such as underwriting, and claim processing.

Furthermore, we select 10 LVLMs for evaluation and conduct a comprehensive analysis of the results. The key findings from the evaluation are as follows: (1) Overall, none of the selected LVLMs score over 70, and LVLMs' performance is not superior to the human baseline results in many tasks, reflecting the complexity and challenge of insurance multimodal tasks; (2) There are significant differences in LVLMs' performance across different insurance types, with better results in auto insurance and health insurance compared to property insurance and agricultural insurance, which indicates that the application of LVLMs in the insurance domain might benefit from a gradual approach; (3) LVLMs exhibit marked differences in performance across different meta-tasks, closely related to the task type and the image type; (4) The gap between open-source and closed-source LVLMs is narrowing, with some open-source models now approaching or even surpassing the capabilities of closed-source models in some tasks; (5) The primary reasons for LVLMs' errors on the INS-MMBench are lack of knowledge and reasoning skills in the insurance field. Although prompt engineering can partially mitigate this issue, further research and optimization specifically for insurance-related tasks are still needed.

In summary, our main contributions are as follow: (1) We introduce INS-MMBench, the first systematic benchmark designed to evaluate LVLMs in the insurance domain; (2) We conduct a

thorough review and distillation of multimodal tasks specific to selected insurance types, using a bottom-up hierarchical task definition methodology; (3) We perform a comprehensive evaluation of representative LVLMs using INS-MMBench, offering insights that guide future advancements of LVLMs in the insurance sector.

## 2 RELATED WORKS

### 2.1 LARGE VISION-LANGUAGE MODELS

With the rapid development of Large Language Models (LLMs) (Chang et al., 2024; Wei et al., 2022; Huang et al., 2022), researchers are leveraging the powerful generalization capabilities of these pre-trained LLMs for processing and understanding multimodal data (Ye et al., 2023; Zhao et al., 2023b; Deshmukh et al., 2023). A key area of focus is the use of Large Vision-Language Models (LVLMs) for visual inputs. LVLMs employ visual encoders and visual-to-language adapters to encode the visual features from image data and align these features with textual features. The combined features are then processed by pre-trained LLMs, leading to significant advancements in visual recognition and understanding (Yin et al., 2023; Wu et al., 2023).

Various open-source and closed-source LVLMs are continuously emerging. In the realm of open-source models, notable examples include LLaMA-Adapter (Zhang et al., 2023), LLaVA (Liu et al., 2024), BLIP-2 (Li et al., 2023c), MiniGPT-4 (Zhu et al., 2023), and InternVL (Chen et al., 2023). These models have successfully integrated visual and textual modalities, achieving commendable results. In the closed-source domain, representative models include GPT-4o (OpenAI, 2024), GPT-4V (Achiam et al., 2023), Gemini (Google, 2024), and Qwen-VL (Team, 2024), all of which have demonstrated outstanding performance in numerous tests and evaluations (Yang et al., 2023; Fu et al., 2023; Li et al., 2023f). We intend to evalute both open-source and closed-source LVLMs to verify the capability of different models in the insurance domain.

### 2.2 BENCHMARKS FOR LARGE VISION-LANGUAGE MODELS

As research into LVLMs intensifies, an increasing number of researchers are proposing benchmarks to evaluate the capabilities of models (Ye et al., 2023; Zhang et al., 2024a; Liu et al., 2023a; Chen et al., 2024b). Based on the scope of capability evaluation, these studies can be categorized into three types: task-specific benchmarks, comprehensive benchmarks, and domain-specific benchmarks.

**Comprehensive benchmarks** are characterized by their breadth and generality. Researchers construct these benchmarks by defining and categorizing the general capabilities and tasks of LVLMs, resulting in a comprehensive and wide-ranging evaluation. Representative studies include LVLM-eHub (Xu et al., 2023), SEED-Bench (Li et al., 2023b;a), MMBench (Liu et al., 2023c), MME, and MMT-Bench (Ying et al., 2024).

**Task-specific benchmarks** focus on particular tasks and types of visual data, providing detailed task definitions. Examples include SciFIBench (Roberts et al., 2024) for scientific images, MMC-Benchmark (Liu et al., 2023b) for charts, MVBench (Li et al., 2023d) (using video frames as input) for videos and SEED-Bench-2-Plus (Li et al., 2024a) for web pages, charts and maps.

**Domain-specific benchmarks** are designed for visual tasks within specific professional domain. Due to the specialized knowledge and unique tasks of these domains, general benchmark cannot fully meet the needs of evaluating LVLMs in these areas. As a result, researchers have begun proposing specialized benchmarks for domains such as healthcare (OmniMedVQA (Hu et al., 2024)), mathematics (Lu et al., 2023; Wang et al., 2024b), autonomous driving (Talk2BEV-Bench (Dewangan et al., 2023)), and geography (Roberts et al., 2023). However, as mentioned previously, the insurance domain and even the finance domain currently lack corresponding domain-specific benchmarks for LVLMs (Chen et al., 2024a; Li et al., 2023e; Lin et al., 2024). Our work introduces INS-MMBench to address this gap, aiming for a significant advancements in the application of LVLMs in the insurance domain.

As shown in Table 1, a thorough comparison is conducted based on the three benchmark categories defined above. Six relevant benchmarks are identified and compared in terms of benchmark type,

Table 1: Comparison of Different Benchmark Datasets.

| Dataset | Type | Size | Models | Potential Overlap |
|---|---|---|---|---|
| INS-MMBench (Ours) | Domain-specific: insurance | 8,856 | 10 | - |
| SEED-Bench (Li et al., 2024a) | Comprehensive | 19,242 | 18 | No |
| MMBench (Liu et al., 2023c) | Comprehensive | 2,974 | 14 | No |
| SciFIBench Roberts et al. (2024) | Task-specific: scientific images | 1,000 | 29 | No |
| MMC-Benchmark Liu et al. (2023b) | Task-specific: charts | 2,000 | 6 | No |
| OmniMedVQA (Hu et al., 2024) | Domain-specific: math | 127,995 | 12 | Yes |
| Mathvista (Lu et al., 2023) | Domain-specific: medical | 5,487 | 9 | No |

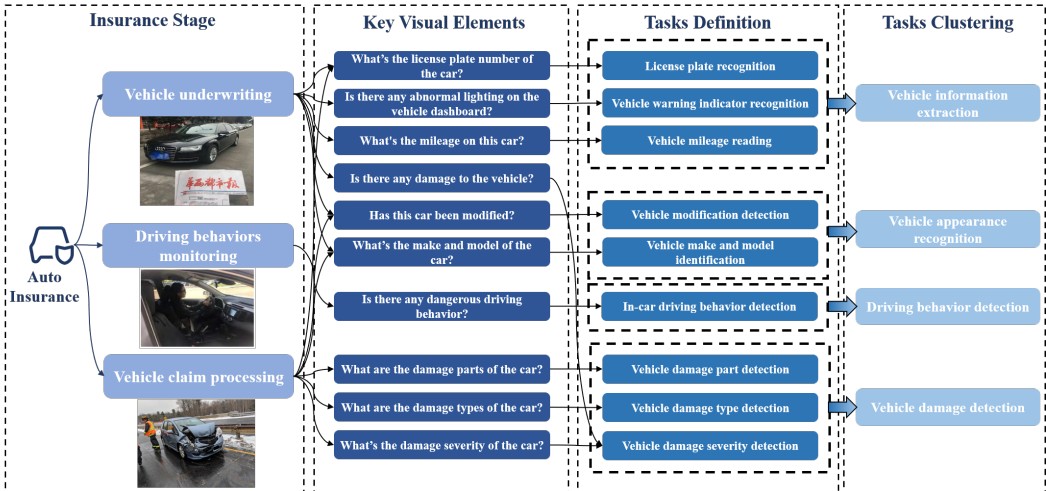

Figure 2: An illustration of our bottom-up hierarchical task definition method. First, we identify and categorize different insurance stages. Next, we enumerate the key visual elements required at each stage. Based on these key visual elements, we define the fundamental tasks. Finally, we cluster the fundamental tasks to form meta-tasks.

dataset size, the number of evaluated models, and potential overlap with our benchmark. This comparison highlights the distinct nature of our benchmark and underscores its contribution to the insurance domain, while also providing context in relation to existing benchmarks across other domains.

## 3 INS-MMBENCH

### 3.1 TASKS

Given the differences in workflows among various types of insurance in practical operations, we select four core types for building this benchmark: auto insurance, commercial/household property insurance, health insurance, and agricultural insurance. Our selection is based on a comprehensive consideration of both the wide coverage these types offer across personal and general insurance, as well as the unique visual tasks associated with each. On the one hand, These categories cover both life and property insurance, which are the most prevalent in the insurance market and highly representative (Weedige et al., 2019; Driver et al., 2018). On the other hand, these insurance types are chosen for their distinct multimodal tasks that are closely aligned with practical applications in the field. For instance, auto insurance involves the assessment of vehicle damage through visual inspection, while property insurance covers evaluations of damaged buildings or personal property.

To ensure that our evaluation tasks closely align with real-world applications in the insurance domain and fully demonstrate the capabilities of LVLMs in this context, we have developed a bottom-up hierarchical task definition methodology. Using this methodology, we construct a systematic visual task framework specifically tailored for the insurance sector. As an example, we discuss the

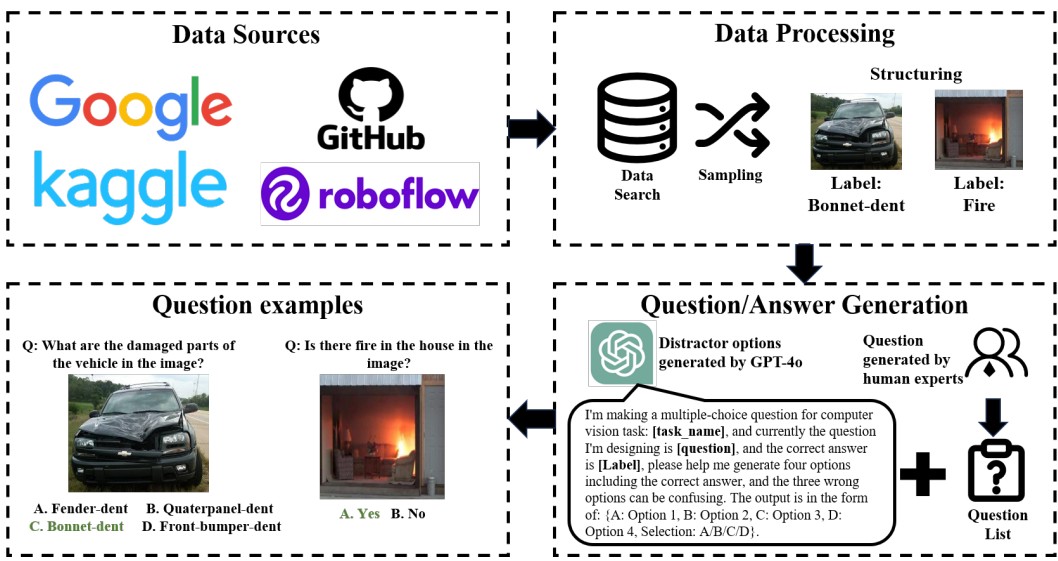

Figure 3: An illustration of our data collection and benchmark construction process. First, we collect datasets from multiple public sources. Next, we perform manual filtering and random sampling of the datasets, followed by the necessary data processing. Finally, both manual effort and GPT-4o are utilized to construct task questions and multiple-choice options, creating a multi-choice visual question dataset.

detailed task construction process for auto insurance (see Figure 2). Initially, based on the insurance value chain theory (Eling & Lehmann, 2018; Eling et al., 2022), we select three key stages rich in multimodal data and tasks: vehicle underwriting, vehicle risk monitoring, and vehicle claim processing. At each stage, we identify the key visual elements that insurance operators need to extract. For instance, during the vehicle underwriting stage, operators must confirm elements such as license plate information, vehicle model, dashboard readings, and vehicle condition, which are crucial for information collection, condition verification, and underwriting decision-making. Further, based on these key visual elements, we define the fundamental tasks. For example, the need to extract license plate information led to the definition of the License Plate Recognition task, while the need to monitor risky driving behavior resulted in the In-car Driving Driving Behavior Detection task. By following this process, we define a total of nine fundamental tasks for auto insurance. Finally, we cluster these fundamental tasks based on their characteristics, forming four meta-tasks. Through this approach, we have constructed a comprehensive set of 12 meta-tasks and 22 fundamental tasks across the four types of insurance.

## 3.2 DATASET COLLECTION

Once the task definition is complete, we start collecting data and constructing the multi-choice visual questions. Our data collection and benchmark construction process (see Figure 3) is as follows:

**Data sources.**  We search for datasets using keywords related to the fundamental tasks in several popular data sources, including Google, Kaggle, Github, and Roboflow. For tasks where multiple public datasets are available, we compare and select these datasets according to usage metrics and user reviews. We select datasets with high adaptability and usability for insurance scenarios, as detailed in Table 2.

**Data processing.**  This stage involves two key subtasks: data sampling and data structuring.

- **Sampling**: We employ a carefully considered sampling methodology. For classification tasks such as Vehicle Damage Severity Detection and Crop Type Identification, where the dataset contains a limited number of labels, we use stratified sampling to ensure balanced

Table 2: An overview of the datasets used in INS-MMBench.

| Insurance type | Meta-tasks | Fundamental tasks | Dataset | Size |
|---|---|---|---|---|
| Auto insurance | Vehicle information extraction | License plate recognition | CCPD (Xu et al., 2018), mjdfodf-qmbuf (workspace, 2023) | 250 |
| | | Vehicle mileage reading | TRODO (Mouheb et al., 2021) | 500 |
| | | Vehicle warning indicator recognition | dataset_dashboard (Dashboarddataset, 2024) | 500 |
| | Vehicle appearance recognition | Vehicle make and model identification | Stanford Cars (Krause et al., 2013) | 500 |
| | | Vehicle modification detection | tuning-car-detection (f-rid nagiyev, 2023) | 100 |
| | Driving behavior detection | Incar driving behavior detection | Driver-Distraction-Dataset (Ezzouhri et al., 2021) | 500 |
| | Vehicle damage detection | Vehicle damage part detection | car_dent_scratch_detection-1 (Sindhu, 2022) | 500 |
| | | Vehicle damage type detection | Cardd (Wang et al., 2023) | 500 |
| | | Vehicle damage severity detection | car-crash-severity-detection (ansonlau1325@gmail.com, 2022) | 308 |
| Property insurance | Property risk assessment | Roof condition assessment | damages-svll3 (Capstone2, 2022) | 500 |
| | | Workplace risk assessment | worker-safety (computer vision, 2022) | 100 |
| | Property anomaly detection | House fire detection | fire-detection-cta61 (College, 2023) | 498 |
| | Property damage detection | House damage type detection | damage-type (Agyemang, 2022) | 469 |
| | | House damage level detection | damage-level (Agyemang, 2021) | 409 |
| Health insurance | Health risk monitoring | Fall detection | Fall Detection Dataset (KANDAGATLA, 2022) | 374 |
| | | Health device reading | blood-pressure-monitor-display (Project, 2024) | 100 |
| | Medical image recognition | Medical image organ recognition | VQA-Med 2019 (Abacha et al., 2019) | 500 |
| | | Medical image abnormality recognition | VQA-Med 2019 (Abacha et al., 2019) | 500 |
| Agricultural insurance | Crop type identification | Field image crop type identification | agricultural crop images (AMAN2000JAISWAL, 2021) | 250 |
| | | Satellite image crop type identification | Drone Imagery Classification Training Dataset for Crop Types in Rwanda (Chew et al., 2020) | 498 |
| | Crop growth status identification | crop growth stage recognition | wheat-growth-stage-challenge (DUTTA, 2023) | 500 |
| | Farmland damage detection | Farmland damage type detection | agriculture-vision (Chiu et al., 2020) | 500 |

representation across labels, minimizing bias. For tasks with more varied outputs, such as Vehicle Plate Recognition, we adopt a random sampling strategy to capture a broad spectrum of responses. Considering the balance of samples and the costs associated with LVLM testing, we set our sample size as the larger of 500 or the maximum number that can be sampled from each fundamental task dataset based on the sampling methodology proposed above. The sample size of each task is shown in Table 2.

- **Structuring**: Label extraction varies depending on the dataset, generally falling into three categories: (1) labels stored in a JSON file, (2) images categorized into folders by label, and (3) labels embedded within image filenames. We process these accordingly, producing a CSV file containing image filenames and their corresponding labels for further use.

**Question and answer construction.** We craft questions for each task, drawing on our designed insurance scenarios. For datasets with up to four labels, the options correspond directly to the dataset's categories (*e.g.,* the four levels of Vehicle Damage Severity: no accident, minor damage, moderate damage, and severe damage). For datasets with more complex or freeform responses, we use GPT-4o to generate plausible incorrect options, thus completing our multiple-choice question format.

## 4 EXPERIMENT

### 4.1 EXPERIMENTAL SETTING

**Selected LVLMs.** We select a representative set of 10 LVLMs for our evaluation. This set includes seven closed-source LVLMs: GPT-4o, GPT-4V, GPT-4o-mini, Gemini 1.5 Flash, QwenVLPlus, QwenVLMax, and Claude3V_Haiku as well as three open-source LVLMs including LLaVA, BLIP-2, and Qwen-VL-Chat.

Table 3: Evaluation results of the LVLMs across different insurance types. The values in the table represent the average accuracy. The highest and second-highest results are highlighted in **bold** and underlined, respectively.

| Model | Overall | Auto insurance | Household/commercial property insurance | Health insurance | Agricultural insurance |
|---|---|---|---|---|---|
| GPT-4o | **69.70** | **86.00** | **63.77** | **76.73** | 36.38 |
| Qwen-VL-Max | 65.33 | 80.86 | 61.99 | 70.60 | 33.18 |
| Gemini 1.5 Flash | 64.21 | 79.40 | 60.18 | 70.31 | 32.84 |
| GPT-4V | 62.79 | 77.35 | 60.55 | 70.82 | 29.23 |
| GPT-4o-mini | 60.66 | 77.77 | 58.53 | 63.61 | 25.80 |
| Qwen-VL-Plus | 54.94 | 71.42 | 48.51 | 64.92 | 20.48 |
| Claude3V_Haiku | 48.95 | 59.95 | 49.63 | 59.02 | 17.91 |
| Qwen-VL-Chat | 48.85 | 57.64 | 45.90 | 65.14 | 21.34 |
| LLaVA | 46.99 | 45.47 | 56.82 | 65.25 | 26.26 |
| Human baseline | 60.45 | 62.22 | 60.00 | 75.00 | **42.50** |

**Evaluation methods.** We employ VLMEvalKit, an open-source evaluation toolkit for LVLMs developed by Duan et al. (2024), to conduct our evaluations. This toolkit supports integrated testing of both closed-source and open-source LVLMs and is adaptable to custom benchmark datasets. VLMEvalKit provides two methods for evaluating responses to multi-choice visual questions: exact matching (finding "A", "B", "C", "D" in the output strings) and LLM-based answer extraction which analyzes the answer outputs using a Large Language Model (we use GPT-4o here). These methods help mitigate the issue of uncontrolled free-form content generation by LVLMs. The accuracy metric is used as the evaluation criterion. Additionally, we conduct a human baseline experiment with three graduate students specializing in Insurance. They are asked to answer a subset of 220 questions (10 from each fundamental task) from the benchmark of 8,856 questions.

## 4.2 MAIN RESULTS

Tables 3 and 4 present the evaluation results of LVLMS across various insurance types and meta-tasks, respectively, using random guessing as the baseline. The results are organized into three sections: the first seven rows present the evaluation results of closed-source models, the middle three rows show the evaluation results of open-source models, and the last row provides the human baseline. Based on the results shown in Tables 3 and 4, the following observations can be made.

**GPT-4o leads in performance but highlights the challenges for LVLMs in insurance tasks.** Overall, GPT-4o outperforms all other models, emerging as the top-performing LVLM on the INS-MMBench with a score of 69.70. When compared to the human baseline, most LVLMs do not significantly outperform humans across many insurance types and tasks, underscoring the challenging nature of insurance-related tasks. These observations indicate significant potential for improvement in applying LVLMs within the insurance domain.

**LVLMs show significant variance across different types of insurance.** Experimental results reveal that both open-source and proprietary LVLMs perform better in tasks related to auto insurance and health insurance compared to those involving property and agricultural insurance. For instance, GPT-4o, which exhibits the best performance, scores 86.00 and 76.73 in auto and health insurance tasks respectively; however, its scores drop to 63.77 and 36.38 in property and agricultural insurance tasks, indicating a gap from practical application. Based on these observations, we suggest that the future deployment of LVLMs in the insurance sector should be a progressive process, initially focusing on areas like auto and health insurance where they are most effective.

**LVLMs show significant variance across different meta-tasks.** Experimental results reveal that LVLMs demonstrate considerable performance variability across various meta-tasks, likely influenced by the capability requirements and image characteristics corresponding to each task. Most models excel in tasks like vehicle information extraction (VAE), vehicle appearance recognition (VAR), and health risk monitoring (HRA), which primarily depend on visual element perception and object detection. In contrast, performance dips in more complex tasks such as household/commercial

Table 4: Evaluation results of the LVLMs across different meta-tasks. The values in the table represent the average accuracy. Specifically, **VIE** denotes vehicle information extraction, **VAR** denotes vehicle appearance recognition, **DBD** denotes driving behavior detection, **VDD** denotes vehicle damage detection, **HPAD** denotes household/commercial property anomaly detection, **HPDD** denotes household/commercial property damage detection, **HPRA** denotes household/commercial property risk assessment, **HRM** denotes health risk monitoring, **MIR** denotes medical image recognition, **CGSI** denotes crop growth stage identification, **CTI** denotes crop type identification, **FDD** denotes farmland damage detection. The highest and second-highest results are highlighted in **bold** and underlined, respectively.

| Model | VIE | VAR | DBD | VDD | HPAD | HPDD | HPRA | HRM | MIR | CGSI | CTI | FDD |
|---|---|---|---|---|---|---|---|---|---|---|---|---|
| GPT-4o | **81.12** | **98.50** | 88.60 | 83.94 | **91.16** | **47.04** | 65.50 | **95.72** | 66.50 | 30.80 | **41.31** | 34.60 |
| Qwen-VL-Max | 75.28 | 98.20 | 74.80 | 81.88 | 80.72 | 45.79 | **71.80** | 88.24 | 64.00 | 29.60 | 40.37 | 26.00 |
| Gemini 1.5 Flash | 67.28 | 96.80 | 79.20 | **84.40** | 74.30 | 46.36 | 70.40 | 81.82 | 66.00 | 36.60 | 38.10 | 21.20 |
| GPT-4V | 72.16 | 93.60 | 66.20 | 80.35 | 88.35 | 41.80 | 65.80 | 94.12 | 62.10 | 23.60 | 39.17 | 20.00 |
| GPT-4o-mini | 70.24 | 95.20 | 85.80 | 75.23 | 89.56 | 39.75 | 60.60 | 94.39 | 52.10 | 23.80 | 34.36 | 15.00 |
| Qwen-VL-Plus | 63.84 | 96.20 | 69.60 | 69.88 | 57.03 | 39.18 | 56.40 | 86.10 | 57.00 | 15.40 | 25.40 | 18.20 |
| Claude3V_Haiku | 45.76 | 86.8 | 52.40 | 66.13 | 75.10 | 27.90 | 62.40 | 84.49 | 49.50 | 19.80 | 23.53 | 7.60 |
| Qwen-VL-Chat | 44.32 | 94.60 | 59.60 | 55.50 | 59.04 | 30.41 | 60.00 | 80.75 | 59.30 | 15.80 | 30.62 | 13.00 |
| LLaVA | 32.64 | 60.20 | 51.80 | 49.69 | 87.35 | 34.85 | 65.00 | 83.69 | 57.54 | 21.40 | 37.57 | 14.20 |
| Human baseline | 76.67 | 45.00 | **100.00** | 46.67 | 70.00 | 46.67 | 60.00 | 85.00 | 65.00 | **60.00** | 35.00 | **40.00** |

Table 5: Comparison of Different LVLMs. VE, LLM and ToP indicate the visual encoder, backbone large language model and number of total parameters, respectively.

| Model | VE | LLM | ToP | Pre-training data | Size | Visual instruction data | Size |
|---|---|---|---|---|---|---|---|
| Qwen-VL-Chat | ViT-bigG/14 | Qwen-7B | 9.6B | Stage1: LAION-en, LAION-zh, LAION-COCO, DataComp, Coyo, CC12M, CC3M, COCO Stage2: LAION-en &zh, DataComp, Coyo, CC12M &3M, SBU, COCO, In-house Data, GRIT, Visual Genome, RefCOCO, RefCOCO+, RefCOCOg, GQA, VGQA, VQAv2, DVQA, OCR-VQA, DocVQA, TextVQA, ChartQA, AI2D, SynthDoG-en & zh, Common Crawl pdf& HTML | 1.4B | Self Instruction dataset | 350K |
| LLaVA | ViT-L/14 | Vicuna | 7B | CC3M | 595K | LLaVA-Instruction | 158K |
| BLIP-2 | ViT-g/14 | FlanT5-XL | 4B | COCO, Visual Genome, CC3M, CC12M, SBU, LAION400M | 129M | - | - |

property damage detection (HPDD) and crop growth stage identification (CGSI), which demand additional domain-specific knowledge or reasoning abilities. Furthermore, LVLMs generally struggle with tasks involving satellite or drone aerial imagery, including household/commercial property risk assessment (HPRA), crop type identification (CTI), and farmland damage detection (FDD), where unique imaging perspectives and data complexities pose additional challenges.

**Narrowing gap between open-source and closed-source LVLMs.** A comparison of the overall performance of open-source and closed-source LVLMs on INS-MMBench indicates that, while there is still a notable gap between the two, some open-source LVLMs are nearing the performance levels of their closed-source counterparts. This trend suggests that as open-source models grow stronger and domain-specific data becomes more abundant, focusing on training high-performance, domain-specific LVLMs could become a key development strategy in the application of LVLMs within the insurance domain.

**Closed-source LVLMs' performance varies by training data size and methods.** Our analysis (shown in Table 5) reveals that both the scale of training data and the methodologies employed are key factors influencing LVLM performance. Qwen-VL-Chat, trained on a massive dataset (over 1.4 billion images in Stage 1 and more in Stage 2), consistently outperforms models like LLaVA and BLIP-2, which are trained on smaller datasets. Moreover, training methods significantly impact versatility. BLIP-2, lacking instruction fine-tuning, struggles with diverse tasks, while LLaVA's emphasis on fine-tuning with its instruction dataset improves performance in specific tasks but limits broader generalization. Qwen-VL-Chat's balanced approach to pre-training and fine-tuning allows it to excel across a wider range of tasks. This demonstrates that both extensive data and well-structured training are essential for strong, generalizable model performance.

## 4.3 Error analysis and mitigation

To provide further insights into the limitations of LVLMs in the insurance domain, we conduct an in-depth analysis of the errors made by selected models on the INS-MMBench. We examine the error patterns of three models: GPT-4o, Gemini 1.5 Flash, and Qwen-VL-Max, categorizing the errors into four types: perception errors (where LVLMs do not recognize or detect objects or content within the image), lack of insurance knowledge or reasoning ability (where LVLMs can recognize and perceive visual content but lack the necessary insurance knowledge or reasoning skills to correctly answer the question), refusal to answer (where LVLMs decline to respond to questions they deem sensitive or illegal), and failure to follow instructions (where LVLMs do not adhere to the provided instructions, resulting in irrelevant responses).

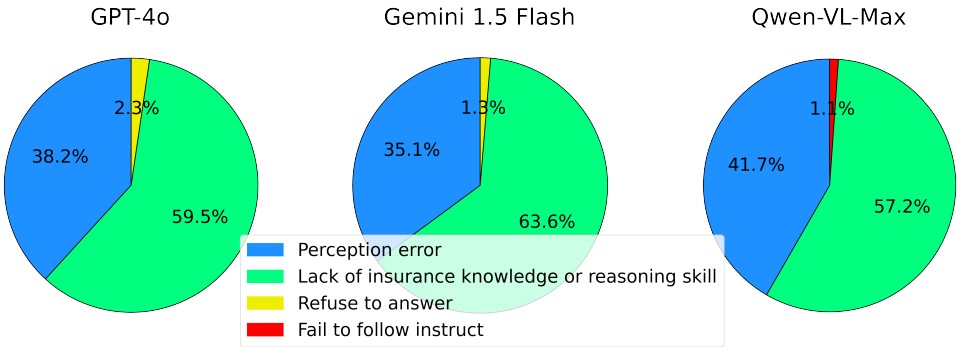

Figure 4: The distribution of error types for GPT-4o, Gemini 1.5 Flash, and Qwen-VL-Chat.

The error analysis results for these models are illustrated in Figure 4. The most common error type is the lack of insurance knowledge or reasoning ability, which accounts for 59.5%, 63.6%, and 57.2% of the errors in GPT-4o, Gemini 1.5 Flash, and Qwen-VL-Max, respectively. Due to insufficient specialized knowledge and analytical skills in the insurance field, LVLMs struggle to accurately assess and judge factors such as risk conditions and the extent of damage. Therefore, optimizing LVLMs for the insurance domain should primarily focus on enriching domain-specific knowledge and enhancing professional capabilities. Perception errors are the second most significant error type. Limited by the capabilities of the visual encoder, LVLMs often fail to fully recognize and capture detailed content in images, leading to misinterpretations. For instance, GPT-4o misidentifies a damaged farmland image as '*an abstract or close-up view of a textured surface with blue and purple hues*'. This type of error is common across LVLMs. Additionally, due to built-in safety monitoring functions, GPT-4o and Gemini 1.5 Flash sometimes incorrectly flag images as illegal and refuse to respond. Qwen-VL-Max, on the other hand, struggles with following instructions, occasionally outputting content in Chinese, which compromises result accuracy.

To address the challenge of insufficient specialized knowledge and analytical skills in the insurance field, we employ prompt engineering as a mitigation method. Specifically, we integrate additional insurance-related information into the original prompts, such as detailed explanations of damage type and severity assessment criteria, to supplement the model's knowledge and support its analytical reasoning. To validate the effectiveness of this approach, we select five models and evaluate them on three tasks that require significant domain expertise in insurance: House Damage Type Detection, Crop Growth Stage Detection, and Vehicle Damage Severity Detection. For each task, we randomly sample 100 instances to create a test set.

As shown in Table 6, the results demonstrate that model performance significantly improves in most cases when enhanced prompts are used. However, in some instances, particularly in the vehicle damage detection tasks for Qwen-VL-Max and Qwen-VL-Plus, the inclusion of additional information leads to confusion when it conflicts with the model's existing reasoning, causing a decline in accuracy. This finding highlights both the effectiveness of prompt engineering as a simple and generalizable method and underscores the need to focus on enhancing LVLMs' specialized knowledge and analytical skills in the insurance domain for further performance improvements.

Table 6: Results of enhanced insurance-related prompts on LVLMs performance across selected tasks. The values represent accuracy (%), and changes in performance are highlighted in green for improvements and red for declines.

| Model | House Damage Type Detection | Crop Growth Stage Detection | Vehicle Damage Severity Detection |
|---|---|---|---|
| GPT-4o | 48.00/**57.00** (+9) | 32.00/**51.00** (+19) | 68.00/**80.00** (+12) |
| GPT-4V | 33.00/**40.00** (+7) | 22.00/**52.00** (+30) | 68.00/**77.00** (+9) |
| Gemini 1.5 Flash | 33.00/**47.00** (+14) | 28.00/**57.00** (+29) | 68.00/**68.00** (-) |
| Qwen-VL-Max | 27.00/**42.00** (+15) | 30.00/**58.00** (+28) | 72.00/**61.00** (-11) |
| Qwen-VL-Plus | 35.00/**38.00** (+3) | 22.00/**60.00** (+38) | 68.00/**58.00** (-10) |

## 5 DISCUSSIONS AND CONCLUSIONS

In this paper, we introduce INS-MMBench, a multimodal benchmark tailored for the insurance domain. To the best of our knowledge, this is the first initiative to systematically review multimodal tasks within this sector and establish a specialized benchmark specifically for it. INS-MMBench comprises 8,856 multiple-choice visual questions, covering four types of insurance, 12 meta-tasks, and 22 fundamental tasks, effectively supporting the assessment of LVLMs' applications in insurance. Additionally, we evaluate several mainstream LVLMs and provide a detailed analysis of the results, offering an initial exploration into the feasibility of employing LVLMs in the insurance sector and providing support for future applications and research directions of LVLMs in this field. We hope our benchmark and findings will guide future research and promote interdisciplinary integration and practical applications within the sector.

However, this study has some limitations. A constraint is the lack of open-source image datasets specific to the insurance domain, primarily due to privacy concerns. The image data utilized in this study, sourced from publicly available datasets, undergoes rigorous curation to ensure that it aligns as closely as possible with real-world insurance application scenarios. Nevertheless, since these images do not from actual insurance cases, there remains an inherent potential for some degree of discrepancy. This issue underscores the need for collaborative efforts between insurance companies and the academic community to develop dedicated open-source image datasets for the insurance domain. Another limitation is that INS-MMBench disaggregates the tasks of LVLMs into various fundamental tasks, assessing LVLM performance from a micro perspective based on task-specific accuracy. In reality, visual tasks in insurance often entail complex integration of multiple capabilities and comprehensive analysis. Addressing this, our next step is to construct a more complex, integrated application benchmark to enable a deeper evaluation of LVLM applications in the insurance domain.

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
