## A    SUPPLEMENTARY MATERIALS

### A.1    ACCESS INSTRUCTIONS

Ins-MMBench and the code for the evaluation experiments will be available.

### A.2    DATA FORMAT

The INS-MMBench dataset is formatted as a multiple-choice visual question and is stored in a .tsv file. Each entry includes the following fields: *index, question, answer, A, B, C, D, image, category, l2 category, and l3 category*. These fields represent the instance index, the question associated with the image, the correct answer, the four answer options (A, B, C, D), the base64-encoded image, the specific type of insurance task, the meta-task name, and the basic task name, respectively.

### A.3    HOSTING PLAN

INS-MMBench will be hosted on Github for the foreseeable future. Should that change, we will create an issue on the GitHub repository.

## B    DATASHEET

### B.1    MOTIVATION

**For what purpose was the dataset created?**

The purpose of creating INS-MMBench was to comprehensively evaluate the capabilities of LVLMs in the insurance field. It includes 12 meta-tasks and 22 fundamental tasks. It is the first comprehensive multimodal benchmark in the insurance field.

### B.2    COMPOSITION

**What do the instances that comprise the dataset represent (e.g., documents, photos, people, countries)?**

The instance of INS-MMBench represents a multiple-choice visual question, including the instance index, the question corresponding to the image, the correct answer, four options A, B, C, and D, the base64 encoding of the image, the insurance type to which the task belongs, the meta-task name, and the basic task name.

**Does the dataset contain all possible instances or is it a sample (not necessarily random) of instances from a larger set?**

Given the expense associated with LVLMs and the need for convenient model evaluation, the image data for each of our core tasks was randomly selected from a larger dataset.

**What data does each instance consist of?**

Each instance in the INS-MMBench dataset includes the following components: *index, question, answer, A, B, C, D, image, category, l2 category, and l3 category*. These components correspond respectively to the instance index, the question related to the image, the correct answer, four answer options (A, B, C, and D), the base64-encoded image, the type of insurance related to the task, the meta-task name, and the basic task name.

**Is there a label or target associated with each instance?**

Yes.

**Is any information missing from individual instances?**

No.

**Are relationships between individual instances made explicit (e.g., users' movie ratings, social network links)?**

N/A.

**Are there recommended data splits (e.g., training, development/validation, testing)?**

No.

**Are there any errors, sources of noise, or redundancies in the dataset?**

The authors are not aware of any errors in the dataset.

**Does the dataset contain data that might be considered confidential (e.g., data that is protected by legal privilege or by doctor-patient confidentiality, data that includes the content of individuals' non-public communications)?**

No.

**Does the dataset contain data that, if viewed directly, might be offensive, insulting, threatening, or might otherwise cause anxiety?**

No.

**Does the dataset relate to people?**

No.

### B.3 COLLECTION PROCESS

**How was the data associated with each instance acquired?**

First, we collected datasets from multiple public sources. Next, we performed manual filtering and random sampling of the datasets, followed by the necessary data processing. Finally, both manual effort and GPT-4o were utilized to construct task questions and multiple-choice options, creating a multi-choice visual question dataset.

**What mechanisms or procedures were used to collect the data (e.g., hardware apparatus or sensor, manual human curation, software program, software API)?**

We used both manual human curation and the GPT-4o API.

**If the dataset is a sample from a larger set, what was the sampling strategy (e.g., deterministic, probabilistic with specific sampling probabilities)?**

We employed the uniform sampling according to different label distributions.

**Who was involved in the data collection process (e.g., students, crowdworkers, contractors) and how were they compensated (e.g., how much were crowdworkers paid)?**

We did not employ external crowdworkers or contractors to construct INS-MMBench.

**Over what timeframe was the data collected?**

We completed data collection in May 2024.

**Were any ethical review processes conducted (e.g., by an institutional review board)?**

No.

### B.4 PREPROCESSING/CLEANING/LABELING

**Was any preprocessing/cleaning/labeling of the data done (e.g., discretization or bucketing, tokenization, part-of-speech tagging, SIFT feature extraction, removal of instances, processing of missing values)?**

Yes, we constructed questions and options for different images using both manual and GPT-4o assistance, and evenly distributed the answers.

### B.5 USES

**Has the dataset been used for any tasks already?**

Yes, we have evaluated 10 LVLMs on this dataset.

**Is there a repository that links to any or all papers or systems that use the dataset?**

No.

**What (other) tasks could the dataset be used for?**

Beyond assessing the capabilities of LVLMs in the insurance sector, this dataset can also serve as fine-tuning training data to enhance LVLM performance.

**Is there anything about the composition of the dataset or the way it was collected and preprocessed/cleaned/labeled that might impact future uses?**

The task scope and number of instances in this dataset are limited. However, future usage and research could expand and enhance based on this foundation.

**Are there tasks for which the dataset should not be used?**

No.

### B.6 DISTRIBUTION

**Will the dataset be distributed to third parties outside of the entity (e.g., company, institution,organization) on behalf of which the dataset was created?**

Yes.

**How will the dataset will be distributed (e.g., tarball on website, API, GitHub)?**

Ins-MMBench will be distributed at the Github repository.

**When will the dataset be distributed?**

The dataset will be made available upon acceptance.

## C    CASES

To offer a detailed view of the task settings in INS-MMBench, we have selected sample cases for each core task and present responses from GPT-4o, Gemini 1.5 Flash, and Qwen-VL-Max in this section.

Figure 5: A sample case of license plate recognition and the responses of GPT-4o, GeminiProVision and Qwen-VL-Max.

## Vehicle milage reading

**Prompt**:

What is the mileage reading of the vehicle in the image?

Options:

A. 119841

B. 118941

C. 119481

D. 191841

**Correct answer**: A

**GPT-4o**:

The mileage reading of the vehicle in the image is 119841.

So, the correct answer is:

A. 119841

**Gemini 1.5 Flash:**

The answer is A. 119841.

**Qwen-VL-Max**:

The mileage reading of the vehicle in the image is option B, 118941 kilometers.

Figure 6: A sample case of vehicle mileage reading and the responses of GPT-4o, GeminiProVision and Qwen-VL-Max.

## Vehicle warning indicator recognition

**Prompt**:

Which indicator icons are lit on the vehicle dashboard in the image?
Options:
A. Door open, Low Fuel
B. Check Engine, High Beam
C. Tire Pressure, Battery
D. Engine Oil, ABS

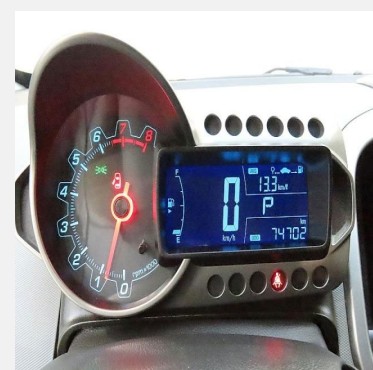

**Correct answer**: A

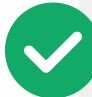

**GPT-4o**:

The correct answer is:
A. Door open, Low Fuel

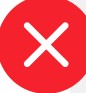

**Gemini 1.5 Flash:**

The correct answer is C. Tire Pressure, Battery.
The image shows two indicator icons lit up on the dashboard: a tire pressure icon and a battery icon.

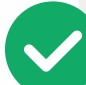

**Qwen-VL-Max**:

The correct answer is A. Door open, Low Fuel. In the image of the vehicle dashboard, there are two indicator icons that are lit up. The first icon indicates that one or more doors are open, and the second icon indicates that the fuel level is low. These symbols serve as important reminders for the driver to ensure safety and proper vehicle maintenance.

Figure 7: A sample case of vehicle warning indicator recognition and the responses of GPT-4o, GeminiProVision and Qwen-VL-Max.

## Vehicle make and model identification

**Prompt**:

What is the make and model of the vehicle in the image?
Options:
A.Jaguar XKR-S Convertible 2012
B. Bentley Continental GT Convertible 2012
C. Rolls-Royce Phantom Drophead Coupe Convertible 2012U
D. Aston Martin DB9 Volante Convertible 2012

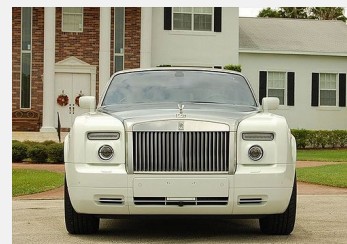

**Correct answer**: C

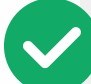

**GPT-4o**:

The vehicle in the image is a Rolls-Royce Phantom Drophead Coupe Convertible 2012. So, the correct answer is:
C. Rolls-Royce Phantom Drophead Coupe Convertible 2012

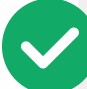

**Gemini 1.5 Flash:**

The answer is C. Rolls-Royce Phantom Drophead Coupe Convertible 2012.

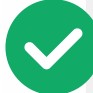

**Qwen-VL-Max**:

The make and model of the vehicle in the image is C. Rolls-Royce Phantom Drophead Coupe Convertible 2012.

Figure 8: A sample case of vehicle make and model identification and the responses of GPT-4o, GeminiProVision and Qwen-VL-Max.

## Vehicle modification detection

**Prompt**:

Please determine whether the vehicle in the image has been modified?
Options:
A. Yes
B. No

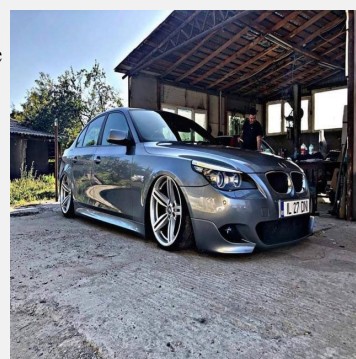

**Correct answer**: A

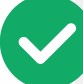

**GPT-4o**:

A. Yes

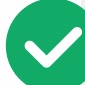

**Gemini 1.5 Flash:**

A. Yes.

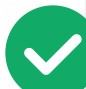

**Qwen-VL-Max**:

A. Yes
The vehicle in the image appears to have been modified with aftermarket wheels and a lowered suspension, which is evident by its stance on the ground. The modifications give the car a distinct look that deviates from its original factory appearance.

Figure 9: A sample case of vehicle modification detection and the responses of GPT-4o, GeminiPro-Vision and Qwen-VL-Max.

## Incar driving behavior detection

**Prompt**:

What is the behavior of the driver in the image?
Options:
A. Texting - left
B. Adjusting mirror - right
C. Eating - right
D. Talking on the phone - left

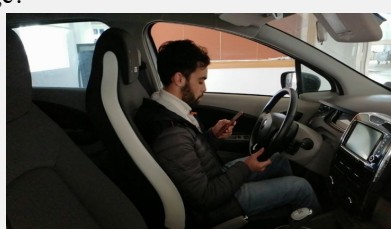

**Correct answer**: A

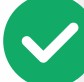

**GPT-4o**:

A. Texting - left

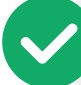

**Gemini 1.5 Flash:**

The answer is A. Texting - left. The driver is holding a phone in his left hand and looking at it.

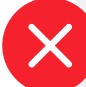

**Qwen-VL-Max**:

The correct answer is D. Talking on the phone - left. In the image, the driver is sitting in the car and holding his phone to his ear, indicating that he is talking on the phone. He is not texting or adjusting the mirror as suggested by options A and B respectively. Option C, eating, does not apply since there is no visible food or action related to eating in the image.

Figure 10: A sample case of incar driving behavior detection and the responses of GPT-4o, Gemini-niProVision and Qwen-VL-Max.

## Vehicle damage part detection

**Prompt**:
What are the damaged parts of the vehicle in the image?
Options:
A. Bonnet-dent
B. Door-scratch
C. Bumper-dent
D. Bonnet-crack

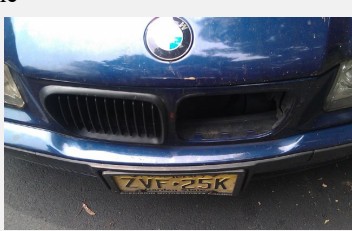

**Correct answer**: A

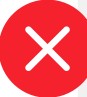
**GPT-4o**:
C. Bumper-dent

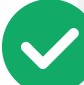
**Gemini 1.5 Flash:**
The answer is A. Bonnet-dent.
The image shows a dent on the bonnet of the vehicle.

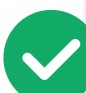
**Qwen-VL-Max**:
The correct answer is A. Bonnet-dent. In the image, there is a blue BMW car with visible damage on its front end. The hood (bonnet) has a dent and some scratches, while one of the grills appears to be missing or broken. The bumper also seems to have some dents and scratches. The license plate holder is bent, and the license plate itself is slightly crooked.

Figure 11: A sample case of vehicle damage part detection and the responses of GPT-4o, GeminiPro-Vision and Qwen-VL-Max.

## Vehicle damage type detection

**Prompt**:

What are the damaged types of the vehicle in the image?

Options:
A. Broken Headlight
B. Dented Door
C. Scratched Paint
D. Tire Flat

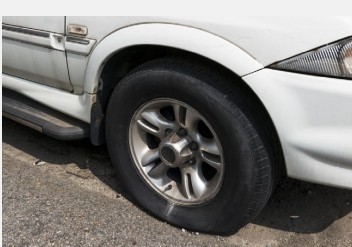

**Correct answer**: D

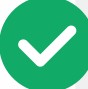 **GPT-4o**:

The correct answer is:
D. Tire Flat

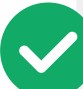 **Gemini 1.5 Flash:**

The answer is D. Tire Flat. The tire is clearly deflated and the car is resting on the rim.

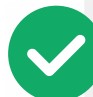 **Qwen-VL-Max**:

The correct answer is D. Tire Flat. In the image, there is a white car with a flat tire on the front left side. The tire appears to be completely deflated and rests against the curb. There is no visible damage to the headlight or door, nor any noticeable scratches on the paint.
Therefore, the only visible damage to the vehicle in this image is the flat tire.

Figure 12: A sample case of vehicle damage type detection and the responses of GPT-4o, GeminiPro-Vision and Qwen-VL-Max.

## Vehicle damage severity detection

**Prompt**:
What is the damaged severity level of the vehicle in the image?
Options:
A. severe
B. moderate
C. no_accident
D. minor

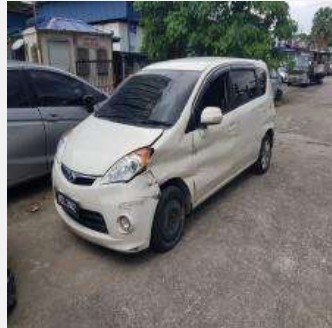

**Correct answer**: D

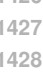

**GPT-4o**:
The correct answer is:
D. minor
The vehicle in the image has some visible damage, particularly around the front left fender and headlight area, but it does not appear to be severe.

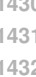

**Gemini 1.5 Flash:**
The answer is B. moderate.
The vehicle has visible damage to the front bumper and fender, but the overall structure of the vehicle appears intact. This suggests moderate damage.

**Qwen-VL-Max**:
The correct answer is D. minor. Based on the image, there appears to be some damage to the front bumper and possibly the headlights of the white car. However, the overall damage seems to be relatively minor compared to more severe accidents or collisions. Therefore, option D, minor, would be the most appropriate description for the damage severity level of the vehicle in the image.

Figure 13: A sample case of vehicle damage severity detection and the responses of GPT-4o, GeminiProVision and Qwen-VL-Max.

## Roof condition assessment

**Prompt**:
What is the condition of the roof of the house in the image?
Options:
A. Damage
B. Not damage

**Correct answer**: A

**GPT-4o**:
 A. Damage

**Gemini 1.5 Flash:**
 A. Damage

**Qwen-VL-Max:**
 A. Damage

Figure 14: A sample case of roof condition assessment and the responses of GPT-4o, GeminiProVision and Qwen-VL-Max.

## Workplace risk assessment

**Prompt**:

What's the risk condition of the workplace in the image?
Options:
A. There is risk of not wearing helmet in the image
B. There is risk of not wearing helmet and vest in the image
C. There is risk of not wearing vest in the image
D. There is no risk in the image

**Correct answer**: C

**GPT-4o**:

A. There is risk of not wearing helmet in the image

**Gemini 1.5 Flash:**

There is risk of not wearing vest in the image

**Qwen-VL-Max:**

C. There is risk of not wearing vest in the image.
In the image, one worker is seen wearing a hard hat and helping another worker who appears to be injured. However, neither of them is wearing a safety vest, which poses a risk on the construction site. Therefore, option C is the correct answer.

Figure 15: A sample case of workplace risk assessment and the responses of GPT-4o, GeminiProVision and Qwen-VL-Max.

## House fire detection

**Prompt**:
Is there fire in the house in the image?
Options:
A. no fire
B. fire

**Correct answer**: B

❌ **GPT-4o**:
A. no fire

❌ **Gemini 1.5 Flash:**
A. no fire

❌ **Qwen-VL-Max**:
Option A: No fire.
There is no evidence of a fire in the image. The room appears to be well-lit and there are no signs of smoke or flames.

Figure 16: A sample case of hosue fire detection and the responses of GPT-4o, GeminiProVision and Qwen-VL-Max.

## House damage type detection

**Prompt**:

What type of damage is done to the house in the image?
Options:
A. Combined Damage
B. Flexural
C. Shear

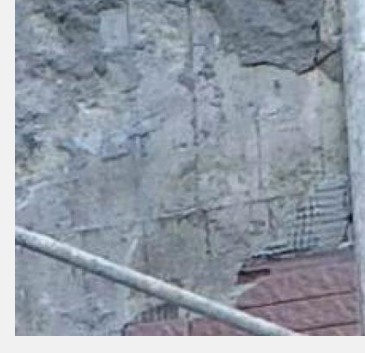

**Correct answer**: A

**GPT-4o**:

The damage shown in the image appears to be characterized by diagonal cracks and a pattern that suggests a sliding or shearing motion. This type of damage is typically associated with shear forces. The correct answer is:
C. Shear

**Gemini 1.5 Flash:**

The answer is A. Combined Damage.
The image shows cracks in the wall, which are a sign of both flexural and shear damage. Flexural damage occurs when a material is bent or deformed, while shear damage occurs when a material is subjected to a force that is parallel to its surface. In this case, the cracks are likely caused by a combination of these two types of damage.

**Qwen-VL-Max**:

The correct answer is A. Combined Damage.\n\nBased on the image provided, it appears that there is combined damage to the house's exterior wall. The wall shows signs of cracking and peeling, which could be due to a combination of factors such as age, weathering, or structural issues. The presence of a metal mesh suggests that repairs have been attempted, but the damage still remains visible. Therefore, option A, Combined Damage, best describes the type of damage seen in the image.

Figure 17: A sample case of hosue damage type detection and the responses of GPT-4o, GeminiPro-Vision and Qwen-VL-Max.

## House damage level detection

**Prompt**:

What is the level of damage to the house in the image?
Options:
A. Undamage
B. Minor Damage
C. Moderate Damage
D. Heavy Damage

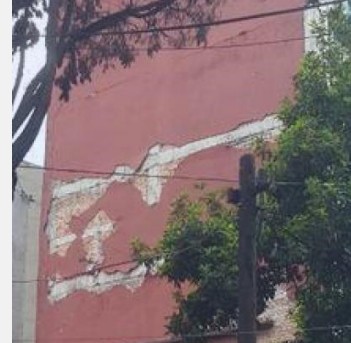

**Correct answer**: C

**GPT-4o**:

The correct answer is:
C. Moderate Damage
The image shows significant cracks and peeling on the exterior wall, indicating structural issues that are more than minor but not extensive enough to be classified as heavy damage.

**Gemini 1.5 Flash:**

The answer is C. Moderate Damage.
The image shows a large crack running across the wall of the house, indicating a significant structural issue. However, the house is still standing and the crack doesn't appear to have caused a complete collapse. Therefore, it's not heavy damage, but it's more than just minor damage.

**Qwen-VL-Max**:

The correct answer based on the image provided is:\n\nC. Moderate Damage\n\nThe image shows a pink building with visible cracks and peeling paint, indicating some degree of structural damage or wear over time. The extent of the damage appears to be moderate rather than minor or heavy.

Figure 18: A sample case of hosue damage level detection and the responses of GPT-4o, GeminiPro-Vision and Qwen-VL-Max.

## Fall detection

**Prompt**:
Does the person in the picture fall down?
Options:
A. fallen
B. not fallen

**Correct answer**: A

**GPT-4o**:
 A. fallen

**Gemini 1.5 Flash:**
 A. fallen

**Qwen-VL-Max**:
 B. not fallen

Figure 19: A sample case of fall detection and the responses of GPT-4o, GeminiProVision and Qwen-VL-Max.

## Health device reading

**Prompt**:

What is the reading of the blood pressure monitor in the image?
Options:
A. SYS:130,DIA:70,PULSE:60
B. SYS:110,DIA:65,PULSE:55
C. SYS:120,DIA:75,PULSE:50
D. SYS:120,DIA:67,PULSE:55

**Correct answer**: D

**GPT-4o**:

The correct answer is:
D. SYS:120,DIA:67,PULSE:55

**Gemini 1.5 Flash:**

D. SYS:120,DIA:67,PULSE:55

**Qwen-VL-Max**:

The correct answer is D. SYS:120,DIA:67,PULSE:55.

Figure 20: A sample case of health device reading and the responses of GPT-4o, GeminiProVision and Qwen-VL-Max.

## Medical image abnormality recognition

**Prompt**:

what is the primary abnormality in this image?
Options:
A. Osteomyelitis
B. Osteogenesis Imperfecta
C. Osteoarthritis
D. Osteosarcoma

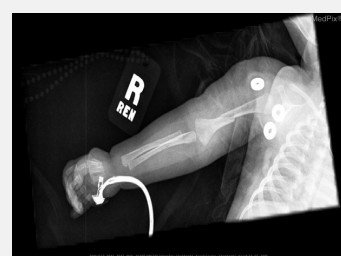

**Correct answer**: **B**

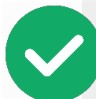

**GPT-4o**:

The primary abnormality in the image is consistent with Osteogenesis Imperfecta (Option B). This condition is characterized by bones that break easily, often with little or no apparent cause. The X-ray shows multiple fractures and bone deformities, which are typical features of Osteogenesis Imperfecta.

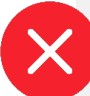

**Gemini 1.5 Flash:**

The answer is D. Osteosarcoma.
Osteosarcoma is a type of bone cancer that typically affects children and young adults. It is characterized by the formation of malignant tumors in the bone. The image shows a large, destructive tumor in the humerus, which is a classic finding of osteosarcoma.

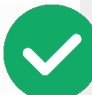

**Qwen-VL-Max**:

B. Osteogenesis Imperfecta

Figure 21: A sample case of medical image abnormality detection and the responses of GPT-4o, GeminiProVision and Qwen-VL-Max.

## Medical image part recognition

**Prompt**:
what is the organ system in this image?
Options:
A. Kidney, Ureter, Bladder
B. Liver, Gallbladder, Pancreas
C. Stomach, Duodenum, Jejunum
D. Lung, Mediastinum, Pleura

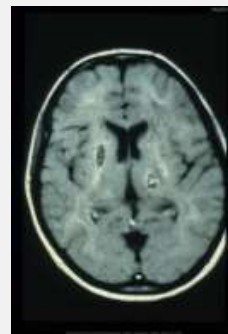

**Correct answer**: D

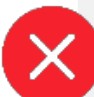

**GPT-4o**:
The organ system in this image is the brain, which is part of the central nervous system. Therefore, none of the provided options are correct.

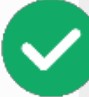

**Gemini 1.5 Flash:**
The answer is D. Lung, Mediastinum, Pleura. This is a brain scan, which is part of the nervous system.

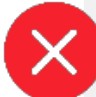

**Qwen-VL-Max**:
The correct answer is B. Liver, Gallbladder, Pancreas. In the provided image, we can see an MRI scan of the liver, gallbladder, and pancreas. The image shows these organs in detail, allowing for evaluation of their structure and potential abnormalities.

Figure 22: A sample case of medical image part recognition and the responses of GPT-4o, Gemi-niProVision and Qwen-VL-Max.

## Field image crop type identification

**Prompt**:
What type of crops are in the image?
Options:
A. Jute
B. Sisal
C. Flax
D. Hemp

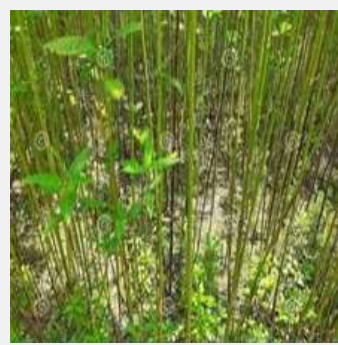

**Correct answer**: A

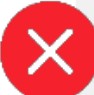 **GPT-4o**:

D. Hemp

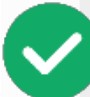 **Gemini 1.5 Flash:**

The answer is A. Jute.
Jute is a natural fiber obtained from the stem of the jute plant. It is a long, soft, and shiny fiber that is used to make a variety of products, including rope, twine, burlap, and canvas. The image shows a field of jute plants, which are characterized by their tall, slender stems and green leaves.

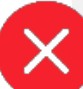 **Qwen-VL-Max**:

The correct answer is C. Flax. The image shows flax plants growing in a field.

Figure 23: A sample case of field image crop type identification and the responses of GPT-4o, GeminiProVision and Qwen-VL-Max.

## Satellite image crop type identification

**Prompt**:
What type of crops are in the image?
Options:
A. Barley
B. Sorghum
C. Wheat
D. Maize

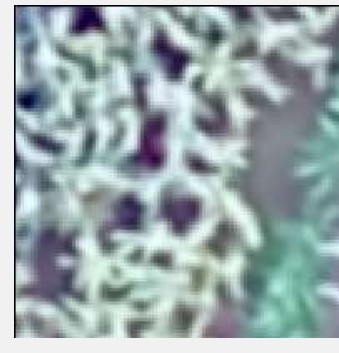

**Correct answer**: D

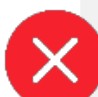

**GPT-4o**:
The image appears to show a close-up of a crop, but the details are not clear enough to definitively identify the type of crop. However, based on the general appearance, it resembles the structure of wheat. So, the correct answer is:
C. Wheat

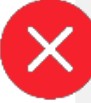

**Gemini 1.5 Flash:**
The answer is C. Wheat.
Wheat crops are characterized by their long, slender leaves and tall stalks. The image shows these features, making it likely that the crops are wheat.

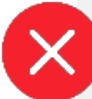

**Qwen-VL-Max**:
Based on the image provided, it is difficult to determine the exact type of crop as the image is blurry and lacks detail. However, based on the given options, the most likely answer would be C. Wheat, as it is a common cereal grain that is often grown in large fields and has a similar appearance to the作物 in the image. However, without more information or a clearer image, it is impossible to确定ly identify the specific type of crop.

Figure 24: A sample case of satellite image crop type identification and the responses of GPT-4o, GeminiProVision and Qwen-VL-Max.

## Crop growth stage recognition

**Prompt**:
What is the growth stage of the crops in the image?
Options:
A. Booting
B. Ripening
C. Tillering
D. Flowering

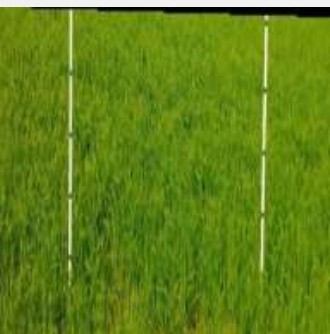

**Correct answer**: A

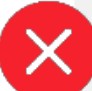 **GPT-4o**:

The growth stage of the crops in the image appears to be "C. Tillering." This stage is characterized by the development of multiple shoots from the base of the plant, which is evident in the dense and lush green appearance of the crop.

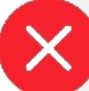 **Gemini 1.5 Flash:**

The answer is C. Tillering.
The crops in the image are in the tillering stage. This is the stage where the plant is producing new shoots from the base of the stem. The shoots are called tillers. The tillering stage is important for the plant because it allows it to produce more leaves and stems, which will eventually produce more grain.

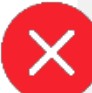 **Qwen-VL-Max**:

Based on the image, it appears that the crops are at the tillering stage, which is when the plants are developing side shoots and branching out to form a dense canopy. Therefore, the correct answer is C. Tillering.

Figure 25: A sample case of crop growth stage recognition and the responses of GPT-4o, GeminiPro-Vision and Qwen-VL-Max.

## Farmland damage type detection

**Prompt**:

What type of damage has occurred on the farmland?
Options:
A. Double plant
B. Lodging
C. Erosion
D. Pest infestation

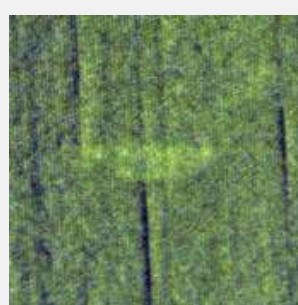

**Correct answer**: A

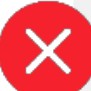

**GPT-4o**:

B. Lodging

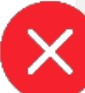

**Gemini 1.5 Flash:**

The answer is C. Erosion.
Erosion is the process of soil and rock being moved from one place to another by wind, water, or ice. The image shows a pattern of vertical lines, which are likely caused by erosion.

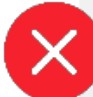

**Qwen-VL-Max**:

Based on the image provided, it appears that there is lodging damage on the farmland. The plants appear to be leaning or fallen over, which is characteristic of lodging damage caused by strong winds or other factors. Therefore, the correct answer is B. Lodging.

Figure 26: A sample case of farmland damage detection and the responses of GPT-4o, GeminiProVision and Qwen-VL-Max.