# OpenReview forum: "INS-MMBench: A Comprehensive Benchmark for Evaluating LVLMs' Performance in Insurance"
_ICLR.cc/2025/Conference — Submitted to ICLR 2025_

### Official Review · Reviewer_XvnZ · 2024-10-27

**Soundness:** 3
**Presentation:** 3
**Contribution:** 3
**Rating:** 6
**Confidence:** 3

**Summary:**

This work introduces INS-MMBench, the first comprehensive benchmark designed to evaluate LVLMs in the insurance domain. INS-MMBench includes four insurance types: auto, property, health, and agriculture, and includes 8856 multiple-choice questions across 12 meta-tasks and 22 fundamental tasks. It is designed to evaluate LVLMs in practical insurance tasks, such as vehicle damage detection and health risk monitoring, combining real-world visual information with insurance-specific questions. Through the experiments, the authors show the current limitations of LVLMs in insurance domain and suggests targeted data and domain knowledge for improving the performance.

**Strengths:**

- The paper is well-written and well-organized.
- This is the first systematic benchmark specifically designed for the LVLMs evaluation in the insurance domain and fills a gap in the current benchmark that often overlooks domain-specific applications.
- The experiments are comprehensive, and thorough error analysis categorizing different types of model failures is provided in the paper.

**Weaknesses:**

- The human baseline experiments only involve 3 graduate students specialized in insurance, which is a small sample size.  This might not accurately represent the range of expertise and variability in the real-world insurance evaluations. I would suggest bringing in more experts from the industry to help perform the human evaluation.
- This work does not discuss potential biases in the data sources or methods for mitigating them, which means that there is a risk that the benchmark may favor certain model behaviors or fail to generalize to different insurance scenarios.

**Questions:**

- How do LVLMs perform on insurance tasks that related to temporal reasoning for example analyzing claim patterns over time? The work evaluates static image understanding, but many insurance tasks require understanding temporal relationships and changes over time, and it seems that those samples are missing from the current benchmark.

---

> ### Author Response · Authors · 2024-11-25
> **Response to Reviewer XvnZ (part 1)**
>
> We sincerely thank you for the thorough review of our work and the insightful feedback. We will address each concern in a detailed, point-by-point manner.
>
> **A1. More human evaluation**
>
> Thank you for highlighting this important issue. We fully agree that involving a broader range of expertise can enhance the representativeness of human evaluation. To address this, in addition to the original evaluations conducted by three graduate students, we have included three industry experts from different leading insurance companies. These experts come from diverse departments, including: Actuarial – Risk Modeling and Analysis, Business Operations – Claims Processing and Customer Interaction, Strategic Development – Insurance Planning and Innovation.This expanded evaluation adds practical insights from different sectors of the insurance industry.
>
> The table below summarizes the updated results, combining the original student evaluations and the assessments from industry professionals:
>
> Table1: Result of human baseline across insurance types.
>
> |             | Overall  | Auto Insurance | Household/Commercial Property Insurance | Health Insurance | Agricultural Insurance |
> |----------------------|----------|----------------|-----------------------------------------|------------------|------------------------|
> | Graduate Students (n=3)       | 60.45 | 62.22       | 60.00         | 75.00          |42.50              |
> | Industry Experts (n=3)       | 67.42 | 75.93       | 64.00         | 74.17          |45.83              |
>
> Table2: Result of human baseline across meta-tasks.
>
> |        | VIE   | VAR   | DBD  | VDD   | HPAD | HPDD | HPRA | HRM  | MIR  | CGSI | CTI  | FDD  |
> |-----------------|-------|-------|------|-------|------|------|------|------|------|------|------|------|
> | Graduate Students (n=3)  | 76.67 | 45.00  | 100.00 | 46.67 | 70.00 | 46.67| 60.00 | 85.00 | 65.00 | 60.00 | 35.00 | 40.00 |
> | Industry Experts (n=3)   | 87.78 | 73.33  | 96.67 | 58.89 | 76.67 | 55.00| 66.67 | 91.67 | 56.67 | 53.33 | 41.67 | 46.67 |

---

> > ### Author Response · Authors · 2024-11-25
> > **Response to Reviewer XvnZ (part 2)**
> >
> > **A2: potential biases and method for mitigating**
> >
> > We appreciate the reviewer’s thoughtful comment regarding potential biases in data sources and the need to address methods for mitigating them. We recognize that biases can arise during dataset creation and that they may influence model behavior or limit generalizability. To address this, we have adopted a structured, multi-stage approach to dataset development, designed to minimize biases at every stage:
> >
> > **Data Retrieval**
> >
> > We retrieved data using a systematic approach guided by task-specific keywords derived from a bottom-up hierarchical task definition method. The benchmark spans 12 meta-tasks and 22 fundamental tasks, including universal challenges such as damage assessment, document analysis, and claims processing, which are not tied to specific legal or regional frameworks. Searches were conducted across platforms such as Kaggle, Google, GitHub, and Roboflow, and datasets were selected based on usage metrics and user reviews. This ensures that our data sources are diverse and reflect the range of tasks in real-world insurance scenarios.
> >
> > The majority of the benchmark is designed to address universal principles and workflows in the insurance domain. For example, tasks like risk assessment and policy matching are broadly applicable across different countries and regulatory environments.
> >
> > **Data Processing**
> >
> > For classification tasks with limited labels (e.g., Vehicle Damage Severity Detection, Crop Type Identification), we employed stratified sampling to ensure balanced representation across labels. This approach minimizes over-representation of dominant labels, reducing potential biases. For tasks with more diverse outputs (e.g., Vehicle Plate Recognition, Vehicle Mileage Reading), we used random sampling to capture a broad spectrum of responses, further reducing the risk of bias stemming from under-represented scenarios.

---

> ### Author Response · Authors · 2024-11-25
> **Response to Reviewer XvnZ (part 3)**
>
> **Question: How do LVLMs perform on insurance tasks that related to temporal reasoning for example analyzing claim patterns over time?**
>
> **A:** Thank you for your valuable feedback and for pointing out the relevance of temporal reasoning in insurance-related tasks. Could the reviewer further clarify the specific meaning of "temporal reasoning" in this context? If it refers to analyzing and studying claim patterns over time, this is primarily a data analytics task in the insurance industry and typically does not involve image data. As such, this falls outside the focus of our current work, which aims to evaluate Large Vision-Language Models (LVLMs) on tasks requiring multimodal reasoning, particularly those that incorporate visual information.
>
> If, on the other hand, "temporal reasoning" implies a multi-step reasoning process, such as making sequential decisions during claim processing, then our supplementary experiment may address this aspect. Specifically, we have added a multi-step reasoning task based on an **auto insurance claim processing scenario**. This task involves input data comprising:
>
> - Images of the damaged vehicle
> - Basic vehicle information (e.g., vehicle value)
> - Insurance details (e.g., policy period and coverage limits)
>
> Using this multimodal input, we instruct the LVLMs to perform multi-step reasoning by sequentially answering five progressively complex questions:
>
> 1. Is the vehicle damaged?
> 2. What is the severity of the damage?
> 3. What is the estimated repair cost for the damage?
> 4. Is the damage eligible for an insurance claim?
> 5. What is the final claimable amount?
>
> Each question requires increasingly sophisticated reasoning and the integration of visual and textual information. For example:
>
> - **Estimating the repair cost** involves assessing the severity of the damage and considering the vehicle's value.
> - **Determining claim eligibility** requires understanding the policy period and comparing it with the incident date.
> - **Calculating the final claimable amount** necessitates reconciling the estimated loss with the insurance coverage limits.
>
> This setup demonstrates the model's capacity for logical reasoning across multimodal inputs in a realistic insurance scenario.
>
> The table below presents the evaluation results of three representative LVLMs. Performance on tasks such as determining vehicle damage, assessing damage severity, and evaluating claim eligibility is measured using **accuracy**, while tasks like loss estimation and final claim decisions are assessed based on the **difference in monetary value** (calculated as the average difference between predicted and actual values).
>
> | Model               | Damage Accuracy | Damage Severity Accuracy | Estimated Loss (Avg. Difference) | Claim Eligibility Accuracy | Final Claim Amount (Avg. Difference) |
> |----------------------|-----------------|---------------------------|-----------------------------------|----------------------------|---------------------------------------|
> | **GPT-4o**          | 82.75%          | 30.50%                   | 6185.61                          | 62.00%                     | 3029.09                               |
> | **QwenVLMax**       | 86.75%          | 29.00%                   | 9584.37                          | 55.50%                     | 3686.32                               |
> | **Gemini-1.5-Flash** | 82.00%          | 25.50%                   | 10894.78                         | 47.50%                     | 4258.76                               |
>
> From the results, we observe the following trends:
>
> 1. **Strong initial performance on visual tasks**: All three models perform reasonably well on the initial task of identifying vehicle damage.
> 2. **Challenges in assessing damage severity**: The models encounter difficulty in accurately assessing the severity of damage, leading to overestimations in loss estimation.
> 3. **Poor reasoning about claim eligibility**: The LVLMs exhibit limited ability to reason about the relationship between the insurance policy's expiration date and claim eligibility.
> 4. **Overestimated final claim decisions**: All models consistently overestimate the final claim amounts compared to actual values.
>
> Among the tested models, **GPT-4o demonstrates relatively better performance**.
>
> This supplementary experiment highlights two key findings. First, current LVLMs face significant challenges in performing complex, multi-step reasoning tasks within insurance workflows. This underscores the need for further development to enhance their applicability to such scenarios. Second, our benchmark's design aligns closely with key business processes in the insurance industry, pinpointing its relevance and applicability to real-world use cases.

---

> ### Author Response · Authors · 2024-11-26
> **Looking forward to the discussion**
>
> Dear Reviewer XvnZ,
>
> We sincerely appreciate the time and effort you have dedicated to reviewing our work, especially given your undoubtedly busy schedule. As the authors-reviewer discussion phase nears its conclusion, we kindly request your attention to our response.
>
> We are eager to understand whether our reply has effectively addressed your concerns and to learn if there are any additional questions or points you would like to discuss.
>
> Thank you once again for your thoughtful consideration, and we look forward to any further feedback you may have.
>
> Best regards,
>
> The Authors

---

> ### Author Response · Authors · 2024-11-28
> **Follow-up on the discussion**
>
> Dear Reviewer XvnZ,
>
> Thank you once again for your time and effort in reviewing our paper and providing constructive feedback. We have strived to prepared the rebuttal based on your feedback and hope we have addressed your concerns effectively. We wonder if there are additional clarifications or details you might need and whether our rebuttal has influenced your score. We are eager to discuss further if necessary.
>
> Best regards,
>
> The Authors

---

> > ### Comment · Reviewer_XvnZ · 2024-11-28
> >
> > Thank you for the response. The authors' replies have addressed my concerns, and I am raising my score based on the clarifications provided.

---

> > > ### Author Response · Authors · 2024-11-29
> > >
> > > We sincerely appreciate the time and effort you dedicated to providing feedback and for raising your score. Your constructive feedback is invaluable in enhancing our study.
> > >
> > > We will incorporate the points addressed in the rebuttal into the revision. As this represents an initial effort to evaluate the performance of Large Vision-Language Models (LVLMs) in the insurance domain, we hope our study will capture the interest of both the research community and industry practitioners, offering meaningful insights to guide the development of future LVLMs.

---

### Official Review · Reviewer_GXCo · 2024-10-27

**Soundness:** 3
**Presentation:** 3
**Contribution:** 2
**Rating:** 6
**Confidence:** 3

**Summary:**

The paper presents a benchmark in the systematic evaluation of LVLMs in this field by introducing INS-MMBench, a domain-specific benchmark designed to assess these models across various insurance-related tasks.

Key contributions:

(1) INS-MMBench is the first comprehensive benchmark tailored for the insurance domain. It covers four representative types of insurance: auto, property, health, and agricultural insurance, reflecting real-world insurance scenarios such as underwriting and claims processing.

(2) The authors used a bottom-up hierarchical task definition approach to identify and refine relevant tasks for each insurance type. They collected and processed datasets to create visual question-answer pairs, ensuring that the benchmark aligns with practical applications in the insurance industry.

(3) The paper evaluates different LVLMs using INS-MMBench. The results highlight the challenges these models face and give some insights.

**Strengths:**

(1) Originality: INS-MMBench is the first benchmark tailored to evaluate LVLMs in the insurance domain. The authors' approach to defining tasks using a bottom-up hierarchical methodology is innovative and ensures that the benchmark aligns with real-world insurance scenarios, making it a pioneering effort in applying LVLMs to this new domain.

(2) Quality: The authors systematically identify and organize multimodal tasks across four types of insurance, and their comprehensive evaluation of ten LVLMs provides some insights. The inclusion of detailed error analysis and the exploration of prompt engineering techniques to mitigate common issues further strengthen the paper, offering practical suggestions for improving model performance.

(3) Clarity: The authors explain each step of their methodology in detail.

(4) Significance: The introduction of INS-MMBench contributes to the field, as it enables a more nuanced evaluation of LVLMs in a domain with substantial practical applications. The benchmark could lead to improved automation in insurance-related tasks, such as claims processing and fraud detection, thus enhancing efficiency and accuracy in the industry. Moreover, by highlighting the narrowing performance gap between open-source and closed-source LVLMs, the paper encourages further research and development, potentially driving advancements in accessible and effective AI solutions for the insurance sector.

**Weaknesses:**

1. Benchmark Definition Lacks Depth in Insurance Scenarios

While INS-MMBench introduces tasks related to insurance, many are more aligned with general, common-sense VQA rather than specialized, nuanced scenarios seen in real-world insurance applications. To better reflect practical needs, the benchmark should include more complex tasks, such as multi-step reasoning or risk assessment based on a mix of visual and contextual data.

2. Overemphasis on Basic Tasks

Some tasks, like license plate recognition, are too basic and can be handled by smaller, specialized models. Evaluating LVLMs on these tasks does not showcase their strengths. Instead, the benchmark should focus on tasks requiring more advanced reasoning, such as verifying claims by cross-referencing multiple data points, to highlight the real capabilities of LVLMs.

3. Limited Emphasis on Reasoning and Higher-Order Tasks

The benchmark lacks tasks that test higher-order reasoning, which is crucial for insurance scenarios. Tasks involving contextual understanding, complex decision-making, and multi-modal integration would better evaluate how well LVLMs can handle real insurance industry challenges.

4. Lack of Focus on Interpretability

Insurance applications require transparency, yet INS-MMBench primarily uses multiple-choice questions, limiting the ability to assess whether models can explain their decisions. Future benchmarks should include tasks that require LVLMs to provide rationale, enabling evaluation of their interpretability, which is critical for building trust in automated systems.

5. Clarification Needed on Table 1
It appears there might be an error in the labeling of the last two rows in Table 1. Currently, "OmniMedVQA" is described as domain-specific for math, and "Mathvista" as domain-specific for medical. Given the names and typical use cases, it seems like these two may have been accidentally switched.

Recommendations:
1. Integrate more complex, domain-specific scenarios that mimic real-world tasks.
2. Replace basic tasks with challenges requiring higher-order reasoning and contextual analysis.
3. Add tasks that require models to explain their answers, enhancing interpretability.

**Questions:**

1. Can you provide more details on how the benchmark tasks were selected?

It would be helpful to understand the criteria used to determine which tasks were included in INS-MMBench. Specifically, how did you ensure that the tasks accurately reflect real-world insurance challenges and not just general visual recognition problems?

2. Have you considered including more complex, multi-step reasoning tasks?

Given the importance of decision-making in insurance, would it be possible to expand the benchmark to include tasks that require multi-modal integration and reasoning (e.g., verifying a claim using images, text descriptions, and numerical data)? This could better showcase the strengths and weaknesses of LVLMs in handling real-world scenarios.

3. How do you envision improving the interpretability of model evaluations?

Since explainability is critical in insurance, have you considered adding tasks that require LVLMs to provide justifications or rationales for their answers? This could allow for a deeper evaluation of how well models understand and explain their decisions, which is crucial for real-world applications.

4. Do you have insights on the performance gap between open-source and closed-source models?

The results indicate a narrowing gap between open and closed-source models. Can you elaborate on specific factors contributing to this trend, and how future benchmarks might encourage more competitive open-source solutions?

---

> ### Author Response · Authors · 2024-11-25
> **Response to Reviewer GXCo (part 1)**
>
> Thank you for thoroughly reviewing our work and providing valuable feedback. We will carefully address each concern in a detailed, and structured manner.
>
> **A1. Benchmark tasks selection**
>
> The selection of tasks in INS-MMBench was guided by a bottom-up hierarchical task definition methodology, which focuses on identifying critical tasks that reflect the operational needs of various insurance scenarios, **rather than general common-sense VQA**.
>
> For example,
>
> **Vehicle Warning Indicator Recognition:** Insurance professionals assess vehicle conditions during claims or underwriting processes. Recognizing specific warning indicators involves understanding automotive systems and their implications for vehicle functionality, safety, and maintenance history.
>
> **Vehicle Modification Detection:** Insurance policies often account for standard vehicle configurations. Detecting modifications requires knowledge of standard vehicle designs and an understanding of risk factors, directly affecting underwriting decisions.
>
> **Vehicle Damage Part, Type, and Severity Detection:** Assessing damage for claims processing involves more than visual recognition; it requires an understanding of vehicle anatomy, repair processes, and cost estimation. For example, detecting structural damage versus cosmetic damage has significant implications for claim payouts and fraud detection.
>
> These distinguish them from general common-sense VQA tasks and underscore the benchmark’s focus on addressing the practical needs of the insurance industry.

---

> > ### Author Response · Authors · 2024-11-25
> > **Response to Reviewer GXCo (part 2)**
> >
> > **A2. Including more complex, multi-step reasoning tasks**
> >
> > Thank you for providing this constructive suggestion. In response, we have added a multi-step reasoning task based on an **auto insurance claim processing scenario**. This task involves input data comprising:
> > - Images of the damaged vehicle
> > - Basic vehicle information (vehicle value)
> > - Insurance details (policy period and coverage limits)
> >
> > Leveraging this combined visual and textual information, we instruct the LVLMs to perform multi-step reasoning by sequentially answering five progressively complex questions:
> >
> > 1. Is the vehicle damaged?
> > 2. What is the severity of the damage?
> > 3. What is the estimated repair cost for the damage?
> > 4. Is the damage eligible for an insurance claim?
> > 5. What is the final claimable amount?
> >
> > Each question requires increasingly sophisticated reasoning and necessitates the integration of visual and textual information. For instance:
> >
> > - **Estimating the repair cost** involves considering the assessed damage severity and the vehicle's value.
> > - **Determining claim eligibility** requires analyzing the policy period.
> > - **Calculating the final claimable amount** involves reconciling the estimated damage cost with the insurance coverage limits.
> >
> > This setup demonstrates the model's capacity for logical reasoning across multimodal inputs in a realistic and practical scenario.
> >
> > The table below presents the results of our evaluation of three representative LVLMs. The performance on tasks such as determining whether the vehicle is damaged, assessing the severity of the damage, and judging claim eligibility is measured using **accuracy**. In contrast, the tasks of loss estimation and final claim decision are evaluated based on the **difference in monetary value** (calculated as the average difference between the predicted and actual amounts).
> >
> > | Model               | Damage Accuracy | Damage Severity Accuracy | Estimated Loss (Avg. Difference) | Claim Eligibility Accuracy | Final Claim Amount (Avg. Difference) |
> > |----------------------|-----------------|---------------------------|-----------------------------------|----------------------------|---------------------------------------|
> > | **GPT-4o**          | 82.75%          | 30.50%                   | 6185.61                          | 62.00%                     | 3029.09                               |
> > | **QwenVLMax**       | 86.75%          | 29.00%                   | 9584.37                          | 55.50%                     | 3686.32                               |
> > | **Gemini-1.5-Flash** | 82.00%          | 25.50%                   | 10894.78                         | 47.50%                     | 4258.76                               |
> >
> >
> > From the results, we observe the following trends:
> >
> > 1. **Strong initial performance on visual tasks**: All three models perform reasonably well on the initial task of identifying vehicle damage.
> > 2. **Challenges in assessing damage severity**: The models encounter varying degrees of difficulty in accurately assessing the severity of the damage, with a notable tendency to overestimate it. This leads to a discrepancy in the estimated loss.
> > 3. **Poor reasoning about claim eligibility**: The LVLMs exhibit poor performance in reasoning about the relationship between the insurance policy's expiration date and claim eligibility.
> > 4. **Overestimated final claim decisions**: The final claim decisions of all tested LVLMs consistently exceed the actual values.
> >
> > Among the three models, **GPT-4o demonstrates relatively better performance**.
> >
> > Through this supplementary experiment, we aim to emphasize two key points. First, current LVLMs perform poorly in complex, multi-step reasoning tasks within insurance scenarios. This highlights a significant challenge and an area for future development to realize LVLMs’ applications in the insurance domain. Second, our benchmark's design of visual tasks closely aligns with critical business processes in insurance, showcasing their relevance and reflecting the potential performance of LVLMs in real-world applications.

---

> > > ### Author Response · Authors · 2024-11-26
> > > **Response to Reviewer GXCo (part 3)**
> > >
> > > **A3. Improving the interpretability of model evaluations**
> > >
> > > Thank you for raising the important point about improving interpretability in model evaluations, particularly in the context of applications like insurance where explainability is critical. To address this, we conducted additional experiments that not only reformulate the questions and answers from a multi-choice format to a free-text format but also require the models to provide justifications or rationales for their answers.
> > >
> > > We manually examined 100 correct responses from each task to assess the accuracy and reliability of the model's reasoning. These tasks include Vehicle Damage Severity Detection, House Damage Level Detection, Medical Image Organ Recognition, and Crop Type Identification Field. Our analysis shows that the model can reasonably and effectively interpret the information in the images and derive answers through logical inference.
> > >
> > > For **vehicle damage severity detection** and **house damage level detection** tasks, the models achieved 100% effectiveness in providing clear and accurate explanations. They successfully identified the damaged parts shown in the images, assessed their impact on core components or functionality, and delivered correct damage evaluations. For example:
> > >
> > > * *"The damage appears to be minor, as it consists of surface scratches and a small dent without any significant structural impact."*
> > >
> > > * *"The damage to the house appears to be severe, with large cracks and exposed areas on the wall, indicating significant structural issues."*
> > >
> > > For **medical image organ recognition** and **crop type identification** tasks, the models provided reasonable and effective explanations in most cases (96%). However, in a small number of instances (4%), they generated speculative or ambiguous responses, which slightly hindered interpretability. For example:
> > >
> > > * *"I can't identify the specific type of crop in the image. However, it appears to be a lush green field, which could indicate a common crop like rice or wheat. For accurate identification, local knowledge or additional context would be needed."*
> > >
> > > Overall, LVLMs demonstrate strong interpretability across these tasks, effectively meeting the interpretability requirements critical to the insurance industry. Their ability to justify predictions enhances their practical applicability and builds trust in automated decision-making systems.
> > >
> > > Looking ahead, we propose that future benchmarks explicitly incorporate tasks requiring models to generate justifications for their outputs. This would enable a more comprehensive evaluation of both accuracy and reasoning capabilities, fostering the development of models that are not only precise but also interpretable. Such advancements are essential for addressing the critical demands of real-world applications in the insurance domain.

---

> > > > ### Comment · Reviewer_GXCo · 2024-11-26
> > > >
> > > > Thanks for your reply.
> > > >
> > > > But, could you explain further on how to evaluate free-text format answers? Additionally, if we want to have better explanation of the multiple choice answers (i.e., give a short reason with the answer together), how would you solve this?

---

> > > > > ### Author Response · Authors · 2024-11-26
> > > > > **Further Response to Reviewer GXCo**
> > > > >
> > > > > We would like to express our sincere gratitude to the reviewer for reading and engaging with our rebuttals.
> > > > >
> > > > > **1. Evaluating Free-Text Answers**
> > > > >
> > > > > In our experiment, we employed a two-stage evaluation approach to ensure the robustness and accuracy of assessing free-text responses:
> > > > >
> > > > > * **Stage 1: Keyword Matching**
> > > > >
> > > > > We initially determined the correctness of the responses by checking if the LVLM-generated answers contained predefined keywords representing the required answer. To account for variations in phrasing, we curated a comprehensive keyword list that included potential synonyms and closely related expressions.
> > > > >
> > > > > * **Stage 2: LLM-Based Verification**
> > > > >
> > > > > For cases where a correct answer was not identified in Stage 1 due to variations beyond keyword coverage, we employed GPT-4o-mini as a verification tool. This model assessed the semantic correctness of the response by comparing it to the ground truth answer and the overall context of the task.
> > > > >
> > > > > **2. Improving Explanations for Multiple-Choice Answers**
> > > > >
> > > > > Regarding the second question, could the reviewer clarify whether the question was about how we believe the explanation of multiple-choice answers could be improved? If so, we think that imposing a structured format on the model’s output would be helpful. This structure would prompt the model to explicitly break down its reasoning process, similar to the chain-of-thought prompting strategy. For example, the model's response could follow this template:
> > > > > ``<Option>, <Key Evidence from the Image>, <Reasoning Process Based on the Visual Evidence>``

---

> > > > > > ### Comment · Reviewer_GXCo · 2024-11-27
> > > > > >
> > > > > > Thanks for your effort. If you could add a case study, that would be better. I will raise my score from 5 to 6.

---

> > > > > > > ### Author Response · Authors · 2024-11-27
> > > > > > >
> > > > > > > We sincerely appreciate your time and effort in providing feedback and thank you for raising your score. The feedback is invaluable in improving our study.
> > > > > > >
> > > > > > > We are currently incorporating the rebuttal and case study into the revised version. As this represents an initial effort to evaluate LVLM performance in the insurance domain, we hope our study will garner attention from both the research community and the industry, offering practical insights into the development of future LVLMs.

---

> ### Author Response · Authors · 2024-11-25
> **Response to Reviewer GXCo (part 4)**
>
> **A4. Insights on the performance gap between open-source and closed-source models**
>
> We observe that the performance gap between open-source and closed-source models is gradually narrowing. This phenomenon may be attributed to the following factors:
>
> **DomainSpecific Task Design:** INS-MMBench includes highly specialized tasks such as license plate recognition, vehicle damage detection, and in-car driving behavior analysis. These tasks provide a level playing field for open-source models by focusing on practical, niche use cases rather than general-purpose vision-language tasks dominated by closed-source models.
>
> **Focus on Real-World Scenarios:** Tasks in INS-MMBench, such as vehicle modification detection or damage severity analysis, require understanding multimodal information in realistic contexts. Open-source models often excel at adapting to such scenarios, narrowing the gap with closed-source counterparts that might focus on broader but less specialized domains.
>
> **Open-Source Community Participation:** Open benchmarks allow the community to test and improve open-source models iteratively, fostering collective progress. This collaborative environment accelerates the pace at which open-source models can close the performance gap.
>
> INS-MMBench can **encourage competitive open-source solutions**. For example, it can encourage participants to fine-tune their models specifically for INS-MMBench tasks. By spotlighting performance gains achieved through fine-tuning, the benchmark can demonstrate the versatility of open-source solutions in adapting to domain-specific challenges.
>
> Additionally, highlighting areas where open-source models excel, such as tasks requiring more domain-specific knowledge (e.g., vehicle damage severity detection), can provide a clear incentive for open-source developers to engage with the benchmark.
>
> Finally, INS-MMBench could feature a "collaborative leaderboard" where participants share improvements and insights, further promoting the open-source ethos of transparency and knowledge sharing.

---

### Official Review · Reviewer_Wwdr · 2024-10-28

**Soundness:** 1
**Presentation:** 2
**Contribution:** 1
**Rating:** 3
**Confidence:** 4

**Summary:**

The potential for Large Vision Language Models (LVLMs) to impact the insurance industry is substantial, yet largely unexplored. This study establishes a benchmark to evaluate LVLM capabilities within the domain, focusing on four main insurance types: auto, property, health, and agriculture. To create the benchmark, the authors gathered multimodal data for each insurance category from public sources and converted it into multiple-choice questions using GPT-4o. They then evaluated popular LVLMs on this benchmark to provide an initial assessment of LVLM performance and reveal current limitations in handling insurance-related content by an error analysis. Finally, the authors try to address gaps in insurance knowledge and reasoning skills by adding insurance-related information to the prompt.

**Strengths:**

1. The motivation behind establishing an insurance benchmark is worthwhile.  Evaluating LVLMs' capabilities on core insurance stages like underwriting and claims processing is practical and meaningful.
2. The benchmark covers a reasonable range of core insurance types relevant to key areas in everyday insurance applications.
3. The study provides an insightful error analysis, highlighting the current limitations of LVLMs in interpreting insurance-specific visual content.

**Weaknesses:**

1. **Misalignment between Intent and Implementation**: While the authors claim the benchmark includes 12 meta-tasks and 22 fundamental tasks across stages like underwriting and claims processing in the Introduction section, the tasks illustrated in the paper are only loosely related to these stages. For example, meta-tasks in auto insurance such as “vehicle information extraction” and “vehicle damage detection” focus heavily on general computer vision tasks rather than directly addressing insurance-specific stages. This makes the benchmark feel more like a vision task set than an insurance task set.
2. **Limited Accessibility for Reproducibility**: Although the authors promise to release the code and dataset, the GitHub repository has not been updated in four months, containing only a readme and a few diagrams. This lack of resources limits my ability to further assess the benchmark’s true rationality and effectiveness.
3. **Limited Novelty**: Some conclusions, such as “performance of closed-source LVLMs varies by training data size and methods,” are too general and widely understood, offering little new insight. The paper would benefit from focusing on more specific findings directly related to the insurance domain.

**Questions:**

Regarding the first limitation, could you share your perspective on how the current selective tasks directly align with the actual stages in the insurance process? For example, specific insurance stages like underwriting or claims processing?

I would consider slightly increasing the score if convinced that the benchmark specifically addresses key insurance stages, rather than being a collection of VQA tasks merely related to the selected insurance categories(auto, property, health, and agriculture).

---

> ### Author Response · Authors · 2024-11-23
> **Reply to Reviewer Wwdr part1**
>
> We greatly appreciate your thorough review of our work and the insightful feedback. We will address the concerns regarding the (1) alignment of our benchmark tasks with key insurance stages, (2) data accessibility, and (3) the novelty of our benchmarks.
>
> **A1. Misalignment Between Intent and Implementation**
>
> Thank you for highlighting this point. Our benchmark design aims to distill essential visual tasks from key stages across various insurance types. The selected tasks can be categorized into **two primary groups**:
>
> **General Tasks Supporting Core Insurance Processes:** Tasks like vehicle information extraction and vehicle appearance recognition are foundational to stages like underwriting and risk assessment. While these may appear as general visual tasks, they are indispensable for the **accurate evaluation of insurance assets**, forming the groundwork for subsequent processes. For instance, extracting vehicle make and model information ensures accurate premium calculations, and vehicle appearance recognition aids in underwriting decision making.
>
> **Insurance-Specific Tasks with Direct Applications:** Examples include damage type detection and damage severity detection, which play pivotal roles in claims processing. These tasks directly inform decisions regarding damage assessment and compensation, ensuring accurate and efficient claim settlements. Such tasks are explicitly designed to mimic real-world insurance workflows, providing a direct bridge between computer visual capabilities and insurance-specific applications.
>
> Additionally, we note that existing works in the domain similarly adopt a multi-module system to extract specific information for different stages of the insurance pipeline. The tasks in our benchmark align with this modular approach, as evidenced by the following works [1,2,3,4]. These examples illustrate that our benchmark aligns closely with real-world applications of visual tasks in the insurance domain. We hope this clarifies how our task selection aligns with key stages like underwriting, claims processing, and risk assessment.
>
> [1] Zhang W, Cheng Y, Guo X, et al. Automatic car damage assessment system: Reading and understanding videos as professional insurance inspectors[C]//Proceedings of the AAAI Conference on Artificial Intelligence. 2020, 34(09): 13646-13647.
>
> [2] Acharya S, Fung G. Mileage Extraction From Odometer Pictures for Automating Auto Insurance Processes[J]. Frontiers in Applied Mathematics and Statistics, 2019, 5: 61.
>
> [3] Kyu, Phyu Mar, and Kuntpong Woraratpanya. "Car damage detection and classification." Proceedings of the 11th international conference on advances in information technology. 2020.
>
> [4] Wang, Fan, et al. "Residential roof condition assessment system using deep learning." Journal of applied remote sensing 12.1 (2018): 016040-016040.
>
> **A2. Limited Accessibility for Reproducibility**
>
> We appreciate your concern regarding the accessibility of our dataset and code. To address this, we have now updated the GitHub repository to include the subset of the benchmark dataset as well as the execution code. This subset includes representative data for each task, ensuring the evaluation of our approach and the rationale behind the task definitions.

---

> > ### Author Response · Authors · 2024-11-23
> > **Reply to Reviewer Wwdr part2**
> >
> > **A3. Limited Novelty**
> >
> > Thank you for raising the concern about some of our conclusions being overly general. We acknowledge that the statement “performance of closed-source LVLMs varies by training data size and methods” may lack a direct explanation of its relevance to the insurance domain. Our intent was to highlight the differences in model architecture and training data across closed-source models.
> >
> > In addition to it, there are **specific findings directly related** to the insurance domain. For instance:
> >
> > **“GPT-4o leads in performance but highlights the challenges for LVLMs in insurance tasks”** underscores that current tasks in the insurance domain present unique challenges for multimodal large vision-language models, emphasizing the complexity and domain specificity required for effective application in this field.
> >
> > The finding **“LVLMs show significant variance across different types of insurance”** reflects that the future deployment of LVLMs in the insurance sector should be a gradual process, initially focusing on domains such as auto and health insurance where their performance is most effective, before expanding into other domains like property or agricultural insurance.
> >
> > Finally, the observation on the **“narrowing gap between open-source and closed-source LVLMs”** highlights a critical insight: the use of open-source models, or even domain-specific fine-tuned models, presents a highly feasible and cost-effective option for insurance companies moving forward.
> >
> > These conclusions aim to underscore the practical implications of our findings, emphasizing their relevance to the unique requirements and potential advancements within the insurance industry.
> >
> > We hope these clarifications and updates address your concerns and demonstrate the alignment of our work with insurance-specific tasks, its accessibility for reproducibility, and its novelty in advancing LVLM applications in the insurance domain. Thank you again for your constructive feedback, and we look forward to any further suggestions you may have.

---

> > ### Comment · Reviewer_Wwdr · 2024-11-23
> >
> > Thank you for your clarification and effort in updating the code. I understand that the visual tasks presented are an important foundational aspect of insurance scenarios, and several tasks serve as direct building blocks for insurance procedures (e.g., damage type detection and damage severity detection). However, they represent only a portion of the capabilities required in the insurance field. The research gap outlined in the paper emphasizes `evaluating the capabilities of LVLMs in insurance`, which goes beyond only foundational visual tasks (such as damage detection) to include processes like writing reports, estimate and claim reviews, policy-based approval workflows, and other key professional procedures.
> >
> > Therefore, focusing solely on visual tasks, while solid in itself, feels too limited to represent the broader scope of the insurance domain. Therefore, this approach does not fully meet my expectations for an insurance-specific benchmark for LVLMs. As a result, I will not increase my score at this time. I agree that the stated research gap in insurance is meaningful, and I believe that if the authors could incorporate more comprehensive insurance-related procedures, this work could become impactful and valuable.

---

> > > ### Author Response · Authors · 2024-11-26
> > > **Further Discussion Regarding the Scope of the Benchmark (part 1)**
> > >
> > > We sincerely appreciate the reviewer’s thoughtful feedback on the scope of our benchmark and are thankful for the opportunity to clarify this concern. The tasks included in INS-MMBench are **not limited to foundational visual tasks** but are specifically designed to address **domain-specific challenges in the insurance field**.
> > >
> > > INS-MMBench includes tasks such as vehicle modification detection, and damage severity detection. These tasks are integral to key insurance workflows, such as claims processing, fraud detection, and risk assessment. They are not general-purpose foundational visual tasks but directly aligned with real-world insurance procedures.
> > >
> > > For example,
> > >
> > > **Vehicle Warning Indicator Recognition:** Insurance professionals assess vehicle conditions during claims or underwriting processes. Recognizing specific warning indicators involves understanding automotive systems and their implications for vehicle functionality, safety, and maintenance history.
> > >
> > > **Vehicle Modification Detection:** Insurance policies often account for standard vehicle configurations. Detecting modifications requires knowledge of standard vehicle designs and an understanding of risk factors, directly affecting underwriting decisions.
> > >
> > > **Vehicle Damage Part, Type, and Severity Detection:** Assessing damage for claims processing involves more than visual recognition; it requires an understanding of vehicle anatomy, repair processes, and cost estimation. For example, detecting structural damage versus cosmetic damage has significant implications for claim payouts and fraud detection.
> > >
> > > We understand that the abstract and introduction could better articulate the domain-specific focus of the benchmark. We propose revising these sections to emphasize that INS-MMBench: (1) evaluates domain-specific multimodal capabilities directly tied to insurance workflows, and (2) establishes a foundation for expanding into higher-level, process-oriented tasks in future iterations.

---

> > > > ### Author Response · Authors · 2024-11-26
> > > > **Further Discussion Regarding the Scope of the Benchmark (part 2)**
> > > >
> > > > We appreciate the reviewer’s perspective and are committed to refining the benchmark to better highlight its insurance-specific scope and future potential. We have added a **multi-step reasoning task** based on an **auto insurance claim processing scenario**. This task involves input data comprising:
> > > > - Images of the damaged vehicle
> > > > - Basic vehicle information (vehicle value)
> > > > - Insurance details (policy period and coverage limits)
> > > >
> > > > Leveraging this combined visual and textual information, we instruct the LVLMs to perform multi-step reasoning by sequentially answering five progressively complex questions:
> > > >
> > > > 1. Is the vehicle damaged?
> > > > 2. What is the severity of the damage?
> > > > 3. What is the estimated repair cost for the damage?
> > > > 4. Is the damage eligible for an insurance claim?
> > > > 5. What is the final claimable amount?
> > > >
> > > > Each question requires increasingly sophisticated reasoning and necessitates the integration of visual and textual information. For instance:
> > > >
> > > > - **Estimating the repair cost** involves considering the assessed damage severity and the vehicle's value.
> > > > - **Determining claim eligibility** requires analyzing the policy period.
> > > > - **Calculating the final claimable amount** involves reconciling the estimated damage cost with the insurance coverage limits.
> > > >
> > > > This setup demonstrates the model's capacity for logical reasoning across multimodal inputs in a realistic and practical scenario.
> > > >
> > > > The table below presents the results of our evaluation of three representative LVLMs. The performance on tasks such as determining whether the vehicle is damaged, assessing the severity of the damage, and judging claim eligibility is measured using **accuracy**. In contrast, the tasks of loss estimation and final claim decision are evaluated based on the **difference in monetary value** (calculated as the average difference between the predicted and actual amounts).
> > > >
> > > > | Model               | Damage Accuracy | Damage Severity Accuracy | Estimated Loss (Avg. Difference) | Claim Eligibility Accuracy | Final Claim Amount (Avg. Difference) |
> > > > |----------------------|-----------------|---------------------------|-----------------------------------|----------------------------|---------------------------------------|
> > > > | **GPT-4o**          | 82.75%          | 30.50%                   | 6185.61                          | 62.00%                     | 3029.09                               |
> > > > | **QwenVLMax**       | 86.75%          | 29.00%                   | 9584.37                          | 55.50%                     | 3686.32                               |
> > > > | **Gemini-1.5-Flash** | 82.00%          | 25.50%                   | 10894.78                         | 47.50%                     | 4258.76                               |
> > > >
> > > >
> > > > From the results, we observe the following trends:
> > > >
> > > > 1. **Strong initial performance on visual tasks**: All three models perform reasonably well on the initial task of identifying vehicle damage.
> > > > 2. **Challenges in assessing damage severity**: The models encounter varying degrees of difficulty in accurately assessing the severity of the damage, with a notable tendency to overestimate it. This leads to a discrepancy in the estimated loss.
> > > > 3. **Poor reasoning about claim eligibility**: The LVLMs exhibit poor performance in reasoning about the relationship between the insurance policy's expiration date and claim eligibility.
> > > > 4. **Overestimated final claim decisions**: The final claim decisions of all tested LVLMs consistently exceed the actual values.
> > > >
> > > > Among the three models, **GPT-4o demonstrates relatively better performance**.
> > > >
> > > > Through this supplementary experiment, we aim to emphasize two key points. First, current LVLMs perform poorly in complex, multi-step reasoning tasks within insurance scenarios. This highlights a significant challenge and an area for future development to realize LVLMs’ applications in the insurance domain. Second, our benchmark's design of visual tasks closely aligns with critical business processes in insurance, showcasing their relevance and reflecting the potential performance of LVLMs in real-world applications.

---

> ### Author Response · Authors · 2024-11-26
> **Looking forward to the discussion**
>
> Dear Reviewer Wwdr,
>
> We sincerely appreciate the time and effort you have dedicated to reviewing our work, especially given your undoubtedly busy schedule. As the authors-reviewer discussion phase nears its conclusion, we kindly request your attention to our response.
>
> We are eager to understand whether our reply has effectively addressed your concerns and to learn if there are any additional questions or points you would like to discuss.
>
> Thank you once again for your thoughtful consideration, and we look forward to any further feedback you may have.
>
> Best regards,
>
> The Authors

---

> ### Author Response · Authors · 2024-11-28
> **Follow-up on the discussion**
>
> Dear Reviewer Wwdr,
>
> Thank you once again for your time and effort in reviewing our paper and providing constructive feedback. We have strived to prepared the rebuttal based on your feedback and hope we have addressed your concerns effectively. We wonder if there are additional clarifications or details you might need and whether our rebuttal has influenced your score. We are eager to discuss further if necessary.
>
> Best regards,
>
> The Authors

---

> > ### Author Response · Authors · 2024-12-01
> > **Follow-Up on Rebuttal (Deadline for Reviewer Feedback Approaching)**
> >
> > Dear Reviewer Wwdr,
> >
> > We hope this message finds you well.
> >
> > As the deadline for reviewer feedback approaches (**December 2**), we kindly ask if you could share whether our rebuttal has adequately addressed your concerns and whether it has influenced your score. If there are any remaining issues, we would greatly appreciate the opportunity to clarify and improve our study.
> >
> > As reviewers ourselves, we deeply understand the importance of constructive and timely engagement with author rebuttals. Your input is invaluable to us, and we are grateful for your time and effort in this process.
> >
> > Best regards,
> >
> > The Authors

---

> > > ### Author Response · Authors · 2024-12-03
> > > **Follow-Up on Rebuttal (10 Hours Remaining for Reviewer Feedback and Questions)**
> > >
> > > Dear Reviewer Wwdr,
> > >
> > > We hope you're doing well.
> > >
> > > With **10 hours** left before the feedback deadline, we’d greatly appreciate your thoughts on whether our rebuttal addressed your concerns and if it has influenced your score. If any issues remain, we’d be glad to clarify further to improve our study.
> > >
> > > Best regards,
> > >
> > > The Authors

---

### Official Review · Reviewer_wQ3N · 2024-11-04

**Soundness:** 3
**Presentation:** 3
**Contribution:** 3
**Rating:** 5
**Confidence:** 5

**Summary:**

The paper introduces INS-MMBench, a comprehensive benchmark designed to evaluate the performance of LVLMs in the insurance domain. It is the first initiative to systematically review multimodal tasks within the insurance sector and establish a specialized benchmark for it.

**Strengths:**

1. Comprehensive Benchmark: The paper presents INS-MMBench, which is the first comprehensive benchmark tailored for evaluating LVLMs in the insurance domain. This benchmark is extensive, covering 8,856 multiple-choice visual questions across 12 meta-tasks and 22 fundamental tasks, providing a robust framework for assessing LVLM capabilities in various insurance scenarios.

2. Systematic Framework: The authors have developed a systematic and hierarchical task definition that ensures the tasks are closely aligned with real-world applications in the insurance industry. This bottom-up approach to task construction enhances the benchmark's relevance and practicality, making it a valuable tool for both research and practical applications.

3. The paper also includes an extensive evaluation of multiple representative LVLMs, offering detailed performance analysis across different insurance types and meta-tasks. This analysis not only validates the effectiveness of the INS-MMBench benchmark but also provides actionable insights into the current capabilities and limitations of LVLMs in the insurance domain, guiding future research and development efforts.

**Weaknesses:**

1. Multi-Choice Format Limitations: This benchmark follows a similar style to MMBench and MME in the general multimodal domain, all of which formulate their questions into multiple-choice formats. While this is an effective method for evaluating model performance, it has limitations that prevent generalization to open-ended question answering, which is more representative of real-world applications.

2. Static Benchmark and Data Leakage: The benchmark is static, which does not mitigate the data leakage problem. This will likely render the benchmark less effective in future developments.

3. Focus on US Insurance Law and Potential Bias: The benchmark primarily focuses on insurance, specifically insurance laws from the United States. This focus may introduce bias into the evaluation process, posing a risk for models developed in different country contexts.

**Questions:**

Check Weaknesses.

**Details Of Ethics Concerns:**

The benchmark primarily focuses on insurance, specifically insurance laws from the United States. This focus may introduce bias into the evaluation process, posing a risk for models developed in different country contexts.

---

> ### Author Response · Authors · 2024-11-25
> **Response to Reviewer wQ3N (part 1)**
>
> We sincerely appreciate the comprehensive review of our work and the thoughtful feedback. We will address the concerns in a point-by-point fashion.
>
> **A1. Multi-Choice Format Limitations**
>
> Thank you for highlighting the potential limitations of the multi-choice format. In response to your feedback, we conducted new experiments based on the original dataset. Specifically, we selected four fundamental tasks: vehicle damage severity detection, medical image organ recognition, house damage level detection, and field crop type identification. While retaining the same images and answers, we reformulated the questions and answers into a free-text format. Evaluation was performed using a comprehensive approach combining keyword matching and LLM-based comparisons.
>
> The experimental results are presented in the table below, where the accuracy of each model is shown as such format (free-text format accuracy/multi-choice format accuracy). We observed that different question-answer formats impacted the models’ performance to varying degrees. These impacts can generally be categorized into two types:
>
> 1. **Tasks with limited-choice classification:** For tasks like vehicle damage severity detection and house damage level detection, where the task inherently involves classification within a finite set of options, the multi-choice format did not introduce additional information. Consequently, the performance changes were within a small and acceptable range of fluctuation.
>
> 2. **Tasks with higher degrees of freedom:** For tasks like medical image organ recognition and crop type identification field, which involve more open-ended responses, the multi-choice format imposed constraints by limiting the selection space. As a result, there was a more pronounced decline in performance when switching to the free-text format. It underscores the importance of developing models better suited to free-text responses, which could inspire future research.
>
> | Model          | Vehicle Damage Severity Detection | House Damage Level Detection | Medical Image Organ Recognition | Crop Type Identification Field |
> |----------------|------------------------------------|------------------------------|---------------------------------|--------------------------------|
> | GPT-4o         | 88.63% / 87.01%                   | 47.43% / 57.95%             | 43.40% / 85.80%                | 71.20% / 90.00%               |
> | QwenVLMax      | 83.44% / 82.14%                   | 41.32% / 56.72%             | 37.20% / 84.00%                | 76.00% / 84.20%               |
> | Gemini-1.5-Flash | 85.71% / 84.09%                  | 62.10% / 57.70%             | 49.60% / 85.60%                | 74.40% / 90.00%               |

---

> > ### Author Response · Authors · 2024-11-25
> > **Response to Reviewer wQ3N (part 2)**
> >
> > **A2. Static Benchmark and Data Leakage**
> >
> > We completely agree that mitigating data leakage and ensuring the benchmark remains dynamic and relevant are essential to its long-term effectiveness. To address this, we have already implemented measures to maintain and expand the dataset dynamically. Specifically:
> >
> > **Regular Data Updates:**
> >
> > We are continuously updating the dataset to incorporate new and diverse multimodal insurance data. The initial version, developed in May, contained 2,200 entries. By October, we expanded the dataset to 8,556 entries, which represents the current version. This update was designed to enhance the dataset's diversity and ensure greater representativeness across various insurance use cases.
> >
> >
> > **Community Collaboration:**
> >
> > We have been actively collaborating with domain experts and the broader research community to drive the dataset's evolution. These experts represent leading insurance companies and come from diverse departments, including Actuarial (risk modeling and analysis), Business Operations (claims processing and customer interaction), and Strategic Development (insurance planning and innovation). By fostering such collaborative contributions, we aim to ensure the dataset remains aligned with emerging trends and addresses the evolving challenges within the insurance domain.
> >
> > **Version Control and Transparency:**
> >
> > We have implemented a systematic versioning strategy to ensure transparency in the dataset's evolution. Each update is documented, detailing changes in data volume, diversity, and structure. This approach enables researchers to track the dataset's development over time and choose the version best suited to their specific needs.
> > These measures demonstrate our commitment to addressing data leakage risks and maintaining the dataset's relevance for future developments. We will continue to prioritize dynamic updates and community engagement to enhance the benchmark's robustness and usability.

---

> > > ### Author Response · Authors · 2024-11-25
> > > **Response to Reviewer wQ3N (part 3)**
> > >
> > > **A3. Focus on US Insurance Law and Potential Bias**
> > >
> > > Thank you for pointing out this important issue. While INS-MMBench is tailored for the insurance domain, it does not exclusively focus on U.S. insurance laws. Instead, the benchmark encompasses a diverse range of tasks, including general insurance concepts, risk analysis, and multimodal reasoning, which are broadly applicable across various international contexts.
> > >
> > > Although regional differences exist, they primarily affect market definitions and regulatory details. For example: In the U.S., insurance is commonly divided into Life/Health Insurance and Property/Casualty Insurance. In countries like the UK and Japan, the industry is classified into Life Insurance and Non-Life Insurance (or General Insurance) [1,2].
> > >
> > > Moreover, the core operational tasks—such as risk assessment, risk monitoring, and claims processing—are highly similar across regions. These tasks form the focus of our benchmark and correspond to established computer vision challenges relevant in diverse contexts partially. Examples include: vehicle damage severity detection [3,6], vehicle damage part detection [4], vehicle mileage reading [5], roof condition assessment [7].
> > >
> > > [1] Cipra T. Financial and insurance formulas[M]. Springer Science & Business Media, 2010.
> > >
> > > [2] https://www.iii.org/sites/default/files/docs/pdf/insurance_factbook_2021.pdf
> > >
> > > [3] Poon F, Zhang Y, Roach J, et al. Modeling and application of neural networks for automotive damage appraisals[J]. SMU Data Science Review, 2021, 5(1): 3.
> > >
> > > [4] Zhang W, Cheng Y, Guo X, et al. Automatic car damage assessment system: Reading and understanding videos as professional insurance inspectors[C]//Proceedings of the AAAI Conference on Artificial Intelligence. 2020, 34(09): 13646-13647.
> > >
> > > [5] Acharya S, Fung G. Mileage Extraction From Odometer Pictures for Automating Auto Insurance Processes[J]. Frontiers in Applied Mathematics and Statistics, 2019, 5: 61.
> > >
> > > [6] Kyu, Phyu Mar, and Kuntpong Woraratpanya. "Car damage detection and classification." Proceedings of the 11th international conference on advances in information technology. 2020.
> > >
> > > [7] Wang, Fan, et al. "Residential roof condition assessment system using deep learning." Journal of applied remote sensing 12.1 (2018): 016040-016040.

---

> ### Author Response · Authors · 2024-11-26
> **Looking forward to the discussion**
>
> Dear Reviewer wQ3N,
>
> We sincerely appreciate the time and effort you have dedicated to reviewing our work, especially given your undoubtedly busy schedule. As the authors-reviewer discussion phase nears its conclusion, we kindly request your attention to our response.
>
> We are eager to understand whether our reply has effectively addressed your concerns and to learn if there are any additional questions or points you would like to discuss.
>
> Thank you once again for your thoughtful consideration, and we look forward to any further feedback you may have.
>
> Best regards,
>
> The Authors

---

> ### Author Response · Authors · 2024-11-28
> **Follow-up on the discussion**
>
> Dear Reviewer wQ3N,
>
> Thank you once again for your time and effort in reviewing our paper and providing constructive feedback. We have strived to prepared the rebuttal based on your feedback and hope we have addressed your concerns effectively. We wonder if there are additional clarifications or details you might need and whether our rebuttal has influenced your score. We are eager to discuss further if necessary.
>
> Best regards,
>
> The Authors

---

> > ### Author Response · Authors · 2024-12-01
> > **Follow-Up on Rebuttal (Deadline for Reviewer Feedback Approaching)**
> >
> > Dear Reviewer wQ3N,
> >
> > We hope this message finds you well.
> >
> > As the deadline for reviewer feedback approaches (**December 2**), we kindly ask if you could share whether our rebuttal has adequately addressed your concerns and whether it has influenced your score. If there are any remaining issues, we would greatly appreciate the opportunity to clarify and improve our study.
> >
> > As reviewers ourselves, we deeply understand the importance of constructive and timely engagement with author rebuttals. Your input is invaluable to us, and we are grateful for your time and effort in this process.
> >
> > Best regards,
> >
> > The Authors

---

> > > ### Author Response · Authors · 2024-12-03
> > > **Follow-Up on Rebuttal (10 Hours Remaining for Reviewer Feedback and Questions)**
> > >
> > > Dear Reviewer wQ3N,
> > >
> > > We hope you're doing well.
> > >
> > > With **10 hours** left before the feedback deadline, we’d greatly appreciate your thoughts on whether our rebuttal addressed your concerns and if it has influenced your score. If any issues remain, we’d be glad to clarify further to improve our study.
> > >
> > > Best regards,
> > >
> > > The Authors

---

### Author Response · Authors · 2024-12-04
**Summary of the Discussion**

Dear Reviewers, AC, and SAC,

We sincerely appreciate the time and effort you have devoted to reviewing our submission and providing thoughtful feedback. At the end of the discussion phase, our submission received scores of 6 (*XvnZ*), 6 (*GXCo*), 3 (*Wwdr*), and 5 (*wQ3N*). While we were unable to receive additional feedback from Reviewers Wwdr and wQ3N during the discussion phase, we hope this summary demonstrates how our rebuttal addressed the concerns raised.

**Responses to Reviewer *wQ3N*:**

Reviewer wQ3N noted that our benchmark, which uses a multiple-choice format like most others, may have limitations in generalizing to open-ended questions. In response, we included an additional experiment in the rebuttal, demonstrating that **our benchmark questions can be easily adapted to an open-ended format**. We are committed to incorporating open-ended questions in the revised version of our work.

Reviewer wQ3N expressed concerns about our benchmark being static and potentially susceptible to data leakage in future developments. We addressed this in the rebuttal by outlining our plan to **regularly update the benchmark** to mitigate these risks.

Reviewer wQ3N also mentioned a perceived focus on US insurance law. In our rebuttal, we clarified that our benchmark is **not limited** to US insurance law and highlighted how the tasks we selected have **broad applicability across various international contexts**, supported by extensive literature.

**Responses to Reviewer *Wwdr*:**

Reviewer Wwdr pointed out that some tasks in our benchmark appeared to be general vision tasks rather than insurance-specific challenges. In the rebuttal, we provided **detailed examples**—such as Vehicle Warning Indicator Recognition, Vehicle Modification Detection, and Vehicle Damage Part, Type, and Severity Detection—to illustrate how these tasks are **not limited** to foundational visual tasks but are specifically designed to address domain-specific challenges in the insurance field.

**Responses to Reviewers *XvnZ* and *GXCo*:**

We addressed concerns raised by Reviewers XvnZ and GXCo by adding more human evaluation, clarifying methods to mitigate potential biases in our data sources, and including experiments on temporal reasoning and multi-step reasoning tasks. We are grateful for the reviewers' constructive discussions and that **our rebuttal were acknowledged**, resulting in raised scores.

As this work represents an ***initial effort*** to evaluate the performance of LVLMs in the insurance domain, we hope it will inspire further research and contribute valuable insights to the development of more effective LVLMs.

Although we were unable to engage in further discussions with some reviewers during the discussion period, we sincerely hope our responses have satisfactorily addressed your concerns.

Thank you once again for your thoughtful reviews and valuable feedback.

Best regards,

The Authors

---

### Meta-Review · Area_Chair_Tay4 · 2024-12-22

**Metareview:**

The paper introduces a new benchmark for multimodal models in insurance, covering auto, property, health, and agriculture. Reviewers liked the domain-specific focus and thorough experiments. However, the tasks are mostly simple visual tasks, missing deeper insurance-specific workflows like multi-step or temporal reasoning. The data appear US-centric, and the approach relies heavily on MCQs which can be limited. These weaknesses reduce its potential impact as a specialized benchmark, and the final recommendation from the AC is reject.

**Additional Comments On Reviewer Discussion:**

During the rebuttal, the authors addressed several concerns in task design, data, and reasoning and evaluation. They added new experiments (e.g., multi-step claims tasks) and input from industry experts. While these changes helped, the benchmark still lacked broad real-world depth.

---

### Decision · Program_Chairs · 2025-01-22

Reject